# The immune suppressive microenvironment affects efficacy of radio-immunotherapy in brain metastasis

Katja Niesel[1], Michael Schulz[1,2], Julian Anthes[1], Tijna Alekseeva[1], Jadranka Macas[3,4],
Anna Salamero-Boix[1,2], Aylin Möckl[1], Timm Oberwahrenbrock[5,6], Marco Lolies[1], Stefan Stein[1],
Karl H Plate[3,4,7], Yvonne Reiss[3,4,7], Franz Rödel[4,7,8] & Lisa Sevenich[1,4,7,*] iD

## Abstract

The tumor microenvironment in brain metastases is characterized by high myeloid cell content associated with immune suppressive and cancer-permissive functions. Moreover, brain metastases induce the recruitment of lymphocytes. Despite their presence, T-cell-directed therapies fail to elicit effective anti-tumor immune responses. Here, we seek to evaluate the applicability of radio-immunotherapy to modulate tumor immunity and overcome inhibitory effects that diminish anti-cancer activity. Radiotherapy-induced immune modulation resulted in an increase in cytotoxic T-cell numbers and prevented the induction of lymphocyte-mediated immune suppression. Radio-immunotherapy led to significantly improved tumor control with prolonged median survival in experimental breast-to-brain metastasis. However, long-term efficacy was not observed. Recurrent brain metastases showed accumulation of blood-borne PD-L1[+] myeloid cells after radio-immunotherapy indicating the establishment of an immune suppressive environment to counteract re-activated T-cell responses. This finding was further supported by transcriptional analyses indicating a crucial role for monocyte-derived macrophages in mediating immune suppression and regulating T-cell function. Therefore, selective targeting of immune suppressive functions of myeloid cells is expected to be critical for improved therapeutic efficacy of radio-immunotherapy in brain metastases.

**Keywords** brain cancer; checkpoint inhibitors; microglia; tumor microenvironment; tumor-associated macrophages

**Subject Categories** Cancer; Immunology

## Introduction

Immunotherapies that aim to reactivate immune responses against tumor cells have been implemented as standard therapies for different cancer entities (Pardoll, 2012). A major focus has been directed to strategies that block immune checkpoint molecules (Havel *et al*, 2019). Response rates to immune checkpoint blockade (ICB) depend on the mutational load of individual cancer types and the immune contexture of the respective tumor microenvironment (McGranahan *et al*, 2016; Mandal *et al*, 2019; Zhao *et al*, 2019). Preclinical and clinical studies revealed that hypermutated cancers with high infiltration of lymphoid effector cells show better response rates than immune excluded tumors with low mutational burden (Nishino *et al*, 2017; Zappasodi *et al*, 2018). Moreover, tumors with high content of immune suppressive cell types such as macrophages and neutrophils show low response rates to ICB (Nakamura & Smyth, 2020). Brain tumors are considered as immunologically cold tumors given the immune privileged status of the central nervous system (CNS) and the highly immune suppressive environment (Antonios *et al*, 2017; Quail & Joyce, 2017; Chongsathidkiet *et al*, 2018; Priego *et al*, 2018; Tomaszewski *et al*, 2019; Friebel *et al*, 2020; Pombo Antunes *et al*, 2020; Schulz *et al*, 2020). T-cell recruitment to brain metastases (BrM) is dependent on the primary tumor entity. Melanoma BrM show high T-cell content, whereas low-to-moderate T-cell recruitment is observed in breast cancer BrM (Harter *et al*, 2015; Berghoff *et al*, 2016; Friebel *et al*, 2020; Klemm *et al*, 2020). Clinical data revealed moderate response rates of ICB applied as monotherapy with melanoma and NSCLC patients showing highest response rates among BrM patients (Brahmer *et al*, 2015; Goldberg *et al*, 2016; Herbst *et al*, 2016; Reck *et al*, 2016; Adams *et al*, 2019). Despite the presence of innate and adaptive immune cell types in brain tumors, there is evidence that anti-tumor

1  Institute for Tumor Biology and Experimental Therapy, Georg-Speyer-Haus, Frankfurt am Main, Germany
2  Biological Sciences, Faculty 15, Goethe University Frankfurt, Frankfurt am Main, Germany
3  Institute of Neurology (Edinger Institute), University Hospital, Goethe University, Frankfurt am Main, Germany
4  Frankfurt Cancer Institute (FCI), Goethe University Frankfurt, Frankfurt am Main, Germany
5  Fraunhofer Institute for Translational Medicine and Pharmacology (ITMP), Frankfurt am Main, Germany
6  Fraunhofer Cluster of Excellence Immune Mediated Diseases (CIMD), Frankfurt am Main, Germany
7  German Cancer Consortium (DKTK), Partner Site Frankfurt/Mainz, Germany and German Cancer Research Center (DKFZ), Heidelberg, Germany
8  Department of Radiotherapy and Oncology, Goethe University Frankfurt, Frankfurt am Main, Germany
   *Corresponding author. Tel: +49 69 63395560; E-mail: sevenich@gsh.uni-frankfurt.de

T-cell responses are inhibited by the highly immune suppressive brain tumor microenvironment (TME) even in the context of ICB (Aslan *et al*, 2020). It was reported that the efficacy of ICB in a melanoma mouse model depends on the presence of extracranial tumors and increased CD8$^+$ T-cell trafficking into BrM (Taggart *et al*, 2018). The necessity of CD8$^+$ T-cell priming and trafficking to CNS lesions to mount anti-tumor immune responses in synergy with ICB was further illustrated by a vascular endothelial growth factor-C (VEGF-C)-mediated modulation of the meningeal lymphatic system (Song *et al*, 2020). This indicates that immune modulation in BrM has the potential to overcome resistance to ICB. Given recent reports on synergistic anti-tumor effects of radio-immunotherapy (Koller *et al*, 2017) and the accumulating evidence that radiation can be used as an immune modulatory agent (Rodriguez-Ruiz *et al*, 2018; Sevenich, 2019; Schulz *et al*, 2020), we sought to evaluate the efficacy of radio-immunotherapy in the syngeneic breast-to-brain metastasis model 99LN-BrM (Bowman *et al*, 2016; Chae *et al*, 2019) with a particular focus on immune modulation induced by whole brain radiotherapy (WBRT) alone and in combination with ICB. Here, we show that WBRT sensitizes BrM to ICB by increasing the relative abundance of CD8$^+$ T cells and by preventing the establishment of lymphoid cell-mediated immune suppressive effects. However, recurrent tumor lesions show accumulation of PD-L1$^+$ monocyte-derived macrophages after radio-immunotherapy suggesting enhanced myeloid-mediated immune suppression to counteract T-cell reactivation in BrM.

# Results

## Cellular composition of the immune compartment in brain metastases

We first characterized the cellular composition of brain metastatic lesions in the syngeneic breast-to-brain metastasis model 99LN-BrM (Appendix Fig S1). Flow cytometric analysis of macrodissected 99LN-BrM revealed that 50% of the cells in 99LN-BrM tumors are CD45$^-$EpCAM$^+$ tumor cells, while the remaining 50% are constituted by different tumor-infiltrating non-cancerous cell types. CD45$^+$ leukocytes represented approximately 20% of the cells of the BrM-associated TME (Fig 1A, Appendix Fig S2). Within the immune compartment, myeloid cells (CD45$^+$CD11b$^+$ cells) represented the most abundant population (Fig 1B). Further classification of BrM-associated myeloid cells (Appendix Fig S2) revealed that brain-resident microglia (MG; CD45$^+$CD11b$^+$Ly6C$^{low}$Ly6G$^-$CD49d$^-$) are the

most prominent population, followed by blood-borne myeloid cells, including monocyte-derived macrophages (MDM; CD45$^+$CD11b$^+$Ly6C$^{low}$Ly6G$^-$CD49d$^+$), inflammatory monocytes (CD45$^+$CD11b$^+$Ly6C$^{high}$Ly6G$^-$), and granulocytes (CD45$^+$CD11b$^+$Ly6C$^{med}$Ly6G$^+$) (Fig 1C). CD11c$^+$MHCII$^+$ cells represented another abundant population. Within this population, conventional dendritic cells 1 (cDC1: CD11c$^+$MHCII$^+$CD11b$^-$CD24$^+$) and 2 (cDC2: CD11c$^+$MHCII$^+$CD11b$^+$) constituted 10 and 50%, respectively. For lymphocytes, we focussed our analysis on CD3$^+$ T cells and B220$^+$ B-cell populations that represented on average 8 and 12.5% of all CD45$^+$ cells (Fig 1C). CD4$^+$ and CD8$^+$ T cells constituted on average 30 and 33% of the CD3$^+$ T-cell population revealing a relatively high amount of double-negative (DN) CD3$^+$ T cells in 99LN-BrM. This population was further stratified into γδ T cells (CD45$^+$CD3$^+$CD4$^-$CD8$^-$γδTCR$^+$), NKT cells (CD45$^+$CD3$^+$CD4$^-$CD8$^-$DX5$^+$), and a remaining DN CD3$^+$ T-cell population (other DN) (Fig 1C). In contrast to previous findings (Chongsathidkiet *et al*, 2018), we did not observe significant differences in CD3$^+$ T-cell numbers in peripheral blood isolated from BrM-bearing mice compared with tumor-free animals (Appendix Fig S3A). However, further stratification of T cells revealed decreased CD8$^+$ T-cell numbers, whereas CD4$^+$ T-cell numbers did not change (Appendix Fig S3B and C).

## Transcription programs in tumor-associated macrophages and lymphocytes in BrM

We next performed RNA sequencing of different FACS purified myeloid and lymphoid populations isolated from brain metastatic lesions, normal brain, or peripheral blood (Dataset EV1). The purity of the sorted populations was validated by the expression of cell type-restricted markers (Fig EV1A and B). We observed pronounced transcriptional changes in tumor-associated cell populations compared with their normal cellular counterparts (Figs 1D and E, and EV1A–C and E, Dataset EV1). The majority of differentially expressed genes (DEG) was restricted to specific cell types. However, we also observed a considerable overlap of common DEG within the analyzed myeloid and lymphoid populations indicating the induction of core transcriptional programs in tumor-associated myeloid and lymphoid populations (Fig EV1D). Consistent with previous findings (Bowman *et al*, 2016; Klemm *et al*, 2020; Schulz *et al*, 2020), we observed loss of expression of microglial markers such as *P2ry12*, *Tmem119*, and *Cx3cr1* in TAM-MG and increased expression of microglial markers in TAM-MDM compared with their normal cellular counterparts (Fig EV1A). The expression of T-cell markers was stable in TILs compared with blood lymphocytes. However, TIL-B cells acquired expression of *Cd3*, *Cd4*, and *Cd8* and showed reduced expression of

**Figure 1. Immune composition of the TME in breast-to-brain metastases.**

A Stacked column depicts proportions of cell types in the TME of 99LN-BrM analyzed by flow cytometry (*n* = 4).
B Stacked column depicts the relative amount of myeloid and lymphoid cells in 99LN-BrM analyzed by flow cytometry (*n* = 5).
C Stacked column depicts the relative amount of myeloid and lymphoid subpopulations in 99LN-BrM based on three flow cytometry panels: myeloid cells (*n* = 5), dendritic cells (*n* = 7), and lymphoid cells (*n* = 6).
D Principal component analysis of tumor-associated myeloid cells vs. blood monocytes and microglia from tumor-free mice (*n* = 3 per condition).
E Principal component analysis of tumor-infiltrating lymphocytes (TIL) vs blood lymphocytes from tumor-free mice (*n* = 3 per condition).
F–I Functional gene annotation of altered cellular pathways in (F) TAM-MG, (G) TAM-MDM, (H) TIL-CD4, and (I) TIL-CD8 compared with control cell types from tumor-free animals. Cutoffs: basemean > 20 and adjusted *P*-value (*Padj*) < 0.05. Adjusted *P*-values were obtained by Wald test and corrected for multiple testing using the Benjamini and Hochberg method. All DEGs based on *Padj* were subjected to analysis.

Source data are available online for this figure.

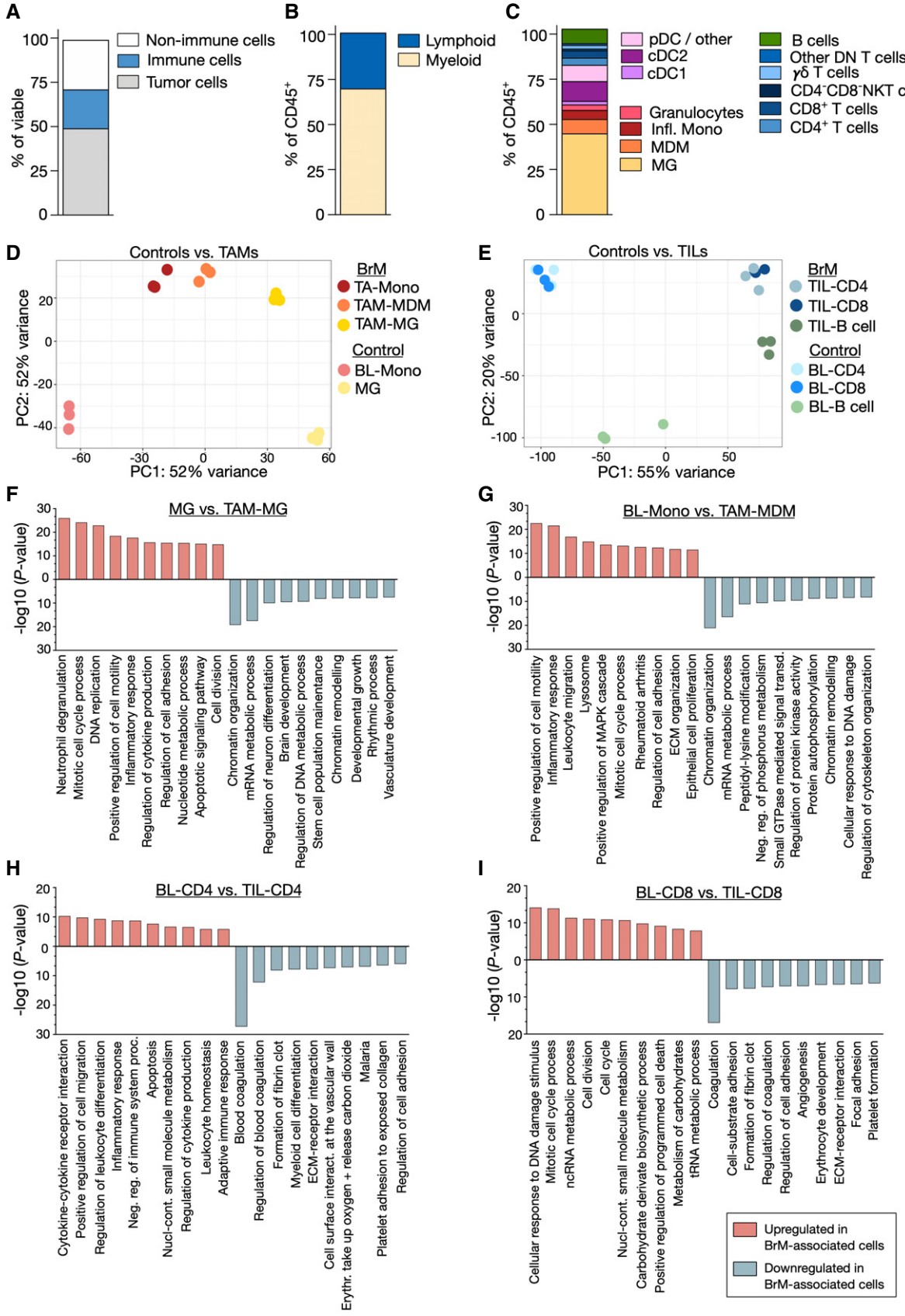

**Figure 1.**

the B-cell markers *Cd19* and *Cd20* (Fig EV1B). Pathway analyses revealed that transcriptional programs in tumor-associated MG were associated with inflammation, cell motility, and proliferation. In contrast, house-keeping functions associated with tissue homeostasis were downregulated in TAM-MG compared with normal MG (Fig 1F). Likewise, TAM-MDM compared with blood monocytes showed induction of inflammatory responses, increased cell motility, and proliferation as well as association to wound repair processes. Moreover, changes in metabolism and signal transductions were observed in TAM-MDM (Fig 1G). Transcriptional programs in CD4[+] T cells indicate the induction of immune responses, increased cell motility and apoptosis (Fig 1H). CD8[+] TILs showed enrichment of pathways associated with proliferation, programmed cell death, and metabolism (Fig 1I). Both, CD4[+] and CD8[+] T cells, showed loss of pathways involved in blood coagulation and platelet formation (Fig 1H and I).

### BrM onset and progression are not dependent on T cells

We employed a T-cell depletion strategy using CD4 and CD8 neutralizing antibodies to analyze the functional role of T cells in controlling BrM onset and progression. Treatment was commenced on day 7 after tumor cell inoculation with doses on three consecutive days followed by weekly injections of the neutralizing antibodies (Fig 2A). Flow cytometry confirmed a significant reduction of T-cell numbers in peripheral blood (Fig 2B and Appendix Fig S4). Moreover, histological assessment revealed depletion of CD3[+] T cells in BrM with no effects on Iba1[+] macrophage numbers (Fig 2C and D). However, T-cell depletion did not affect BrM onset or progression (Fig 2E–G). This finding suggests that TILs are ineffective in controlling disease progression. To address the activation status of tumor-infiltrating T cells, we queried the RNA sequencing data for the expression of a panel of effector and exhaustion markers (Wherry & Kurachi, 2015; Winkler & Bengsch, 2019) and observed prominent acquisition of an exhausted T-cell phenotype in both CD4[+] and CD8[+] T cells in BrM (Fig 2H).

### Blood-borne myeloid cells play a critical role in regulating T-cell activity

We next sought to evaluate which cells within the BrM TME are involved in modulating T-cell activity. Quantitative real-time PCR

(qRT-PCR) indicated high expression levels of PD-L1 in tumor cell lines, whereas PD-1 expression was low (Fig EV2A). Flow cytometry demonstrated that up to 40% of 99LN-BrM cells, 25% of TS1-BrM cells, and 90% of B16-F10 cells express PD-L1 *in vitro* (Fig EV2B). *In vivo*, PD-1 expression was found in lymphocytes, whereas PD-L1 expression was mostly associated with tumor cells (Fig EV2C). Flow cytometry confirmed that 40% of T cells expressed PD-1 with lower PD-1 expression on tumor cells and myeloid cells (Fig 3A). PD-L1 expression was highest on tumor cells followed by myeloid cells and T cells (Fig 3B). PD-L1 expression was almost absent on brain-resident myeloid cells in tumor-free animals but was strongly induced in tumor-associated myeloid cells (Fig 3C). Interestingly, we observed low PD-L1 expression on TAM-MG. In contrast, granulocytes, monocytes, and MDMs showed significantly higher PD-L1 expression (Fig 3D). We next queried the expression of genes involved in antigen presentation as well as co-regulatory factors with stimulatory and/or inhibitory functions in TAMs. This analysis revealed that the majority of genes associated with antigen presentation and regulation of T-cell activity show higher expression in TAM-MDM compared with TAM-MG (Fig 3E). Only the expression of the co-regulatory factor *Cd112* and *Vista* was significantly higher in TAM-MG compared with TAM-MDM. In T cells, *Ctla4* and *Pd-1* were the most prominently expressed co-inhibitory receptors (Fig 3F). The critical role for TAM-MDM in regulating T-cell activity was further supported by functional annotation of enriched pathways in TAM-MG vs. TAM-MDM. Top 10 pathways in TAM-MG were associated with cell migration, angiogenesis, and wound repair mechanism, whereas pathways enriched in TAM-MDM were associated with immune modulation in particular regulation of lymphocyte activity (Fig 3G). We next performed multiplex immunofluorescence staining to evaluate the spatial distribution of TILs in BrM relative to tumor cells and TAMs using marker combinations that allow to discriminate TAM populations based on the expression of the macrophage marker Iba1 and the microglial marker Tmem119. Iba1[+]Tmem119[+] MG were the most abundant population in the adjacent brain parenchyma and in the peri-tumor area. Infiltration of Iba1[+]Tmem119[+] cells into BrM lesions was observed. However, Iba1[+] cells with low or no Tmem119 expression were most prominent within tumor lesions, most likely representing BrM infiltrating TAM-MDM. We observed close proximity of most T cells to tumor cells and

**Figure 2. The effect of T-cell depletion on tumor onset and progression in 99LN-BrM.**

A  Experimental design of T-cell depletion by αCD4 and αCD8a antibody treatment.
B  Representative flow cytometry blots of blood samples from mice of both groups 3 weeks after treatment start showing successful depletion of CD45[+]CD3[+] T cells in the αCD4 + αCD8 treatment group.
C  Representative IHC images of CD3[+] T cells and Iba1[+] macrophages/MG in the isotype and αCD4 + αCD8 group. Scale bar; 100 µm.
D  Quantification of Iba1[+] and CD3[+] cells in IHC sections of BrM from the isotype (*n* = 9) and αCD4 + αCD8 group (*n* = 7).
E  Representative MRI pictures of BrM in early, medium, and late stage of both groups.
F  Kaplan–Meier curves depict BrM-free survival of mice in the isotype and αCD4 + αCD8 group (Isotype *n* = 19, αCD4 + αCD8 *n* = 19).
G  Tumor growth curves depict the increase in absolute BrM volume for each mouse in both groups over time (isotype *n* = 19, αCD4 + αCD8 *n* = 19).
H  Heatmap displays the relative expression level of T-cell effector and exhaustion marker in TIL-CD4 and TIL-CD8 compared with blood lymphocytes. Values display the average per group based on vst values (BL-CD4 or BL-CD8 with *n* = 3 for each group) vs. BrM-associated TIL-CD4 or TIL-CD8 (*n* = 3 for each group).

Data information: Numerical data in (D) are represented as scatter dot plot with line at mean ± SD. *P*-values were obtained by unpaired *t*-test in (D) or based on adjusted *P*-value (Padj) obtained by Wald test and corrected for multiple testing using the Benjamini and Hochberg method in (H) with *P < 0.05, **P < 0.01, and ***P < 0.001. Exact *P*-values can be found in Appendix Table S3 for (D) and the Data Source file associated with Fig 2 for (H).
Source data are available online for this figure.

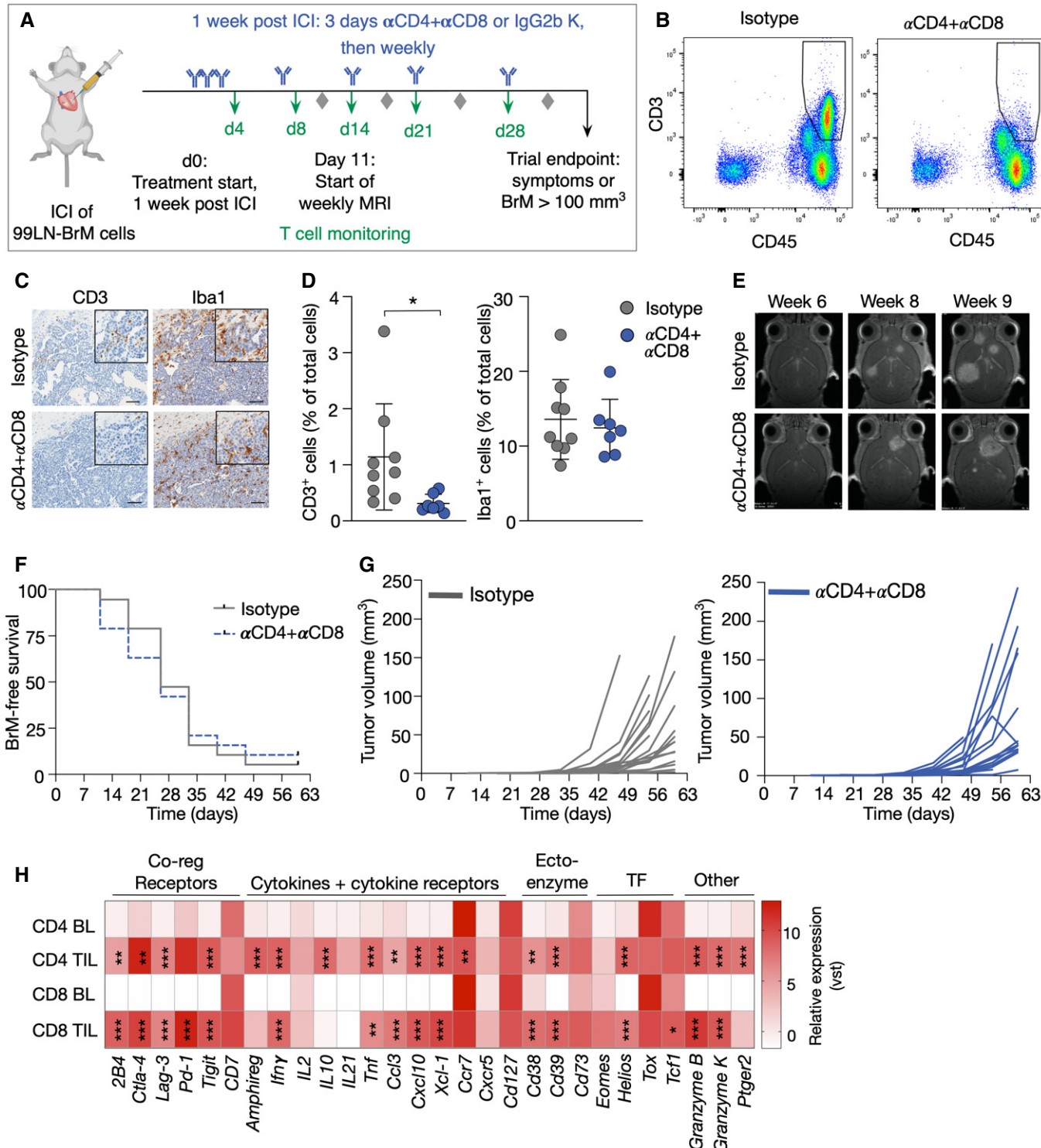

**Figure 2.**

TAMs (Fig 3H and I). Interestingly, CD4+ and CD8+ T cells were localized closer to Iba1+Tmem119− TAMs compared with Iba1+Tmem119+ TAMs (Fig 3H and I) suggesting close proximity of T cells to TAMs associated with modulating T-cell functions in BrM.

**Effects of radiotherapy on the immune cell composition in brain metastases**

Given recent reports on immune modulatory effects of ionizing radiation (IR), we evaluated effects of radiotherapy on the BrM TME. IR

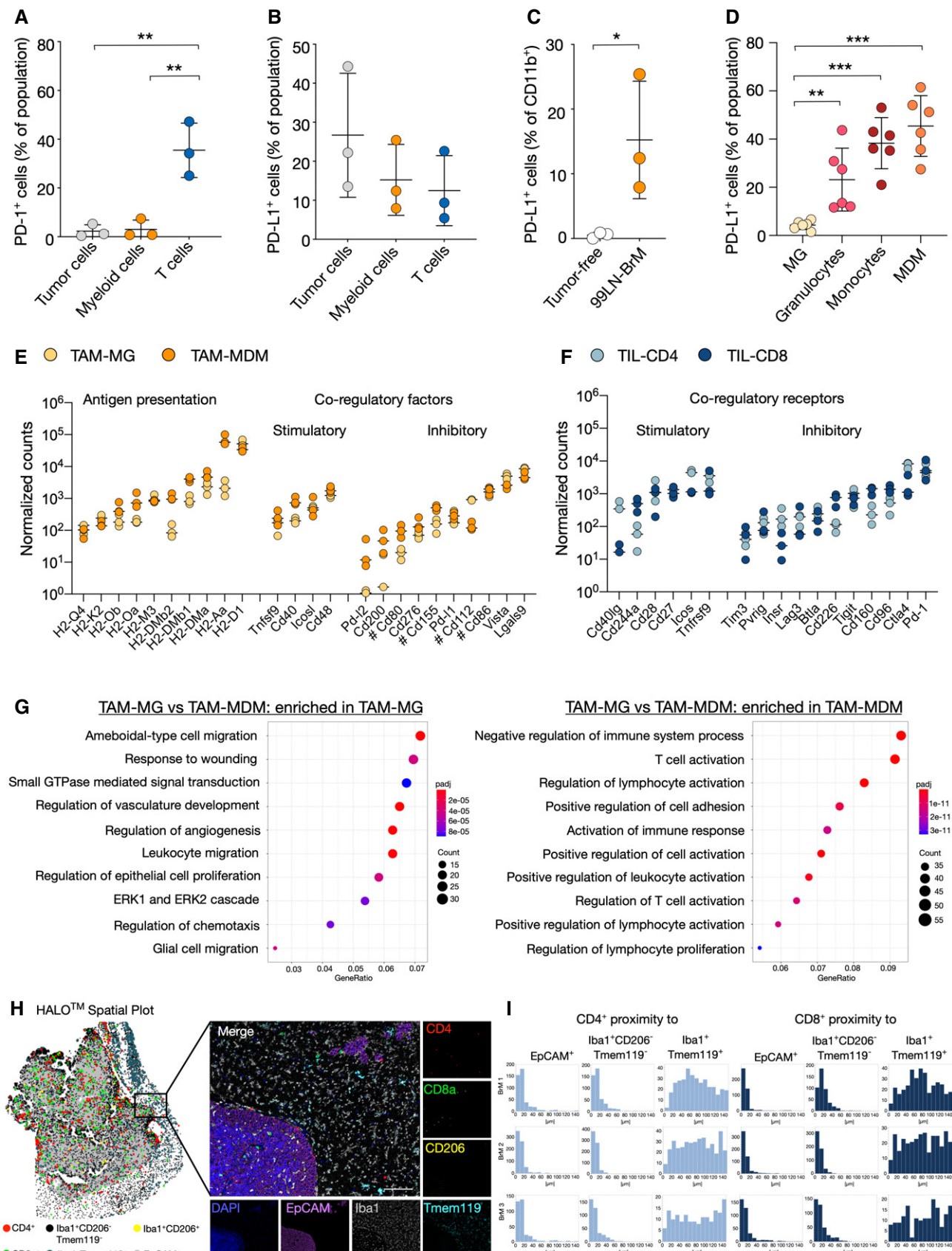

**Figure 3.**

**Figure 3. Expression of immune checkpoints in breast-to-brain brain metastases.**

A    Surface expression of PD-1 *in vivo* in 99LN-BrM quantified by flow cytometry (*n* = 3).
B    PD-L1 in 99LN BrM quantified by flow cytometry (*n* = 3).
C    Flow cytometric analysis of PD-L1 in tumor-free brain and 99LN-BrM (*n* = 3).
D    Flow cytometric analysis of PD-L1 expression on 99LN-BrM-associated myeloid cell types (*n* = 6).
E    Expression of genes associated with antigen presentation and co-regulatory factors in TAM-MG and TAM-MDM. *n* = 3 per condition. Values are depicted as normalized counts. Genes labeled with # can have stimulatory effects depending on the co-regulatory receptor on T cells.
F    Gene expression of co-regulatory receptors in TIL-CD4 and TIL-CD8. *n* = 3 per condition. Values are depicted as normalized counts.
G    Functional gene annotation of altered cellular pathways in TAM-MG vs. TAM-MDM. Left panel depicts pathways enriched in TAM-MG, and right panel depicts pathways enriched in TAM-MDM. Adjusted *P*-value (*Padj*) was obtained by Wald test and corrected for multiple testing using the Benjamini and Hochberg method.
H    Representative HALO spatial plot showing the spatial distribution of cell types with indicated marker combination in 99LN-BrM and representative IF images of multiplexed histology. Scale bar; 100 μm.
I    HALO proximity histogram depicts the distance (μm) between the indicated cell types for individual mice (*n* = 3).

Data information: In (A–D), data are represented as scatter dot plot with lines at mean ± SD. In (E + F), data are represented as scatter dot plot with lines at median. *P*-values were obtained by unpaired *t*-test with \**P* < 0.05, \*\**P* < 0.01, and \*\*\**P* < 0.001. Exact *P*-values can be found in Appendix Table S3.
Source data are available online for this figure.

was applied as fractionated whole brain radiotherapy with 2 Gy on five consecutive days (5 × 2 Gy) in 1 arc using the Small Animal Radiation Research Platform (SARRP) (Wong *et al*, 2008) as previously described (Chae *et al*, 2019). Treatment was commenced when animals developed established metastatic lesions based on magnetic resonance imaging (MRI) (Fig 4A and B). Histological evaluation of the number of TAMs showed no difference in response to 5 × 2 Gy WBRT at trial end point (Fig 4C). Further stratification revealed that WBRT did not affect the ratio of Iba1$^+$Tmem119$^+$ to Iba1$^+$Tmem119$^-$ TAMs (Fig 4D). This was supported by flow cytometry based on CD49d expression levels indicating no change in the amount of TAM-MG and TAM-MDM in response to WBRT (Fig 4E). Moreover, infiltration of other myeloid cell populations was not altered either (Fig 4E). Similar findings were obtained for dendritic cells with no significant changes in DC infiltration and cDC1:cDC2 ratio in response to WBRT (Fig 4F–J). We did not observe significant changes in the overall number of T cells or the amount of Foxp3$^+$ T regulatory cells in BrM after WBRT (Fig 4K–M). However, the number of CD8$^+$ T cells was increased leading to a higher CD8:CD3 T-cell ratio after WBRT (Fig 4K and N), which was further supported by flow cytometry (Fig 4O–R).

## Effects of WBRT on clonal expansion and T-cell receptor repertoires

We next performed TCR sequencing in untreated and irradiated BrM and cervical lymph nodes (CLN) (Fig 5A and Dataset EV2). This analysis further confirmed that the proportion of T cells in BrM and CLN does not change in response to WBRT (Fig 5B). Interestingly, all BrM samples showed a high maximal productive frequency (Figs 5C and EV3A, and Dataset EV2) and high productive clonality (> 0.1) (Fig 5D) suggesting clonal expansion as already indicated by functional gene annotation that revealed an enrichment of pathways associated with proliferation in CD8$^+$ T cells (Fig 1I). WBRT led to a marginal increase in the clonality of TILs in BrM, independent of the template number (Figs 5D and EV3B). Analysis of the clonal space homeostasis led to the identification of hyperexpanded clones in 8 of 9 samples. In some samples, these clones constituted nearly 25 % of the entire T-cell pool (Fig 5E). Large clones were identified in each BrM sample with varying proportions (Fig 5E). However, the proportions were not significantly different between control and WBRT samples (Fig EV3C). Analysis of TCRseq results with Lorenz curves and Gini indexes revealed a skewing of TCR repertoires

**Figure 4. Immune cells in BrM after WBRT.**

A    Experimental design of tumor initiation and application of WBRT.
B    Representative MRI pictures of untreated and irradiated mice at d0 and d14 after treatment start.
C    Quantification of Iba1$^+$ macrophages in IHC sections of 99LN-BrM at trial end point (Ctrl *n* = 27, WBRT *n* = 18).
D    Representative IF images of 99LN-BrM stained for the macrophage marker Iba1 (red) and the MG marker Tmem119 (white). Higher magnification images present areas in the peri-tumor region and in the tumor core. DAPI was used as nuclear counterstain. Scale bar; 100 μm.
E    Quantification of different myeloid populations in 99LN-BrM tumors with and without irradiation by flow cytometry at d14 (Ctrl *n* = 5, WBRT *n* = 4).
F    Representative IF images of 99LN-BrM with and without WBRT stained for DCIR2 (red) and EpCAM (green). DAPI (blue) was used as nuclear counterstain. Scale bar; 100 μm.
G    Quantification of CD11c$^+$MHCII$^+$ immune cells by flow cytometry in 99LN-BrM with and without WBRT.
H, I    Relative proportion of cDC1 and cDC2 in 99LN-BrM with and without WBRT.
J    cDC1:cDC2 ratio in 99LN-BrM based on flow cytometry data (H + I).
K    Representative IHC images of CD3$^+$, FoxP3$^+$, and CD8$^+$ T cells in 99LN-BrM sections with and without WBRT. Scale bar; 100 μm.
L    Quantification of CD3$^+$ cells in IHC sections of 99LN-BrM. (Ctrl *n* = 26, WBRT *n* = 18).
M    FoxP3:CD3 ratio in IHC sections of 99LN-BrM. (Ctrl *n* = 25, WBRT *n* = 18).
N    CD8:CD3 ratio in IHC sections of 99LN-BrM. (Ctrl *n* = 25, WBRT *n* = 18).
O–Q    Quantification of CD3$^+$, CD4$^+$, and CD8$^+$ T cells in 99LN-BrM tumors with and without irradiation by flow cytometry.
R    CD4:CD8 ratio in 99LN-BrM based on data (P + Q).

Data information: In (G–J), *n* = 6 for control and *n* = 7 for WBRT. In (O–R), *n* = 10 for control and for WBRT. In (C + E + G–J + L–N, O–R), data are represented as scattered dot plot with line mean ± SD. *P*-values were obtained by unpaired *t*-test with \**P* < 0.05. Exact *P*-values can be found in Appendix Table S3.
Source data are available online for this figure.

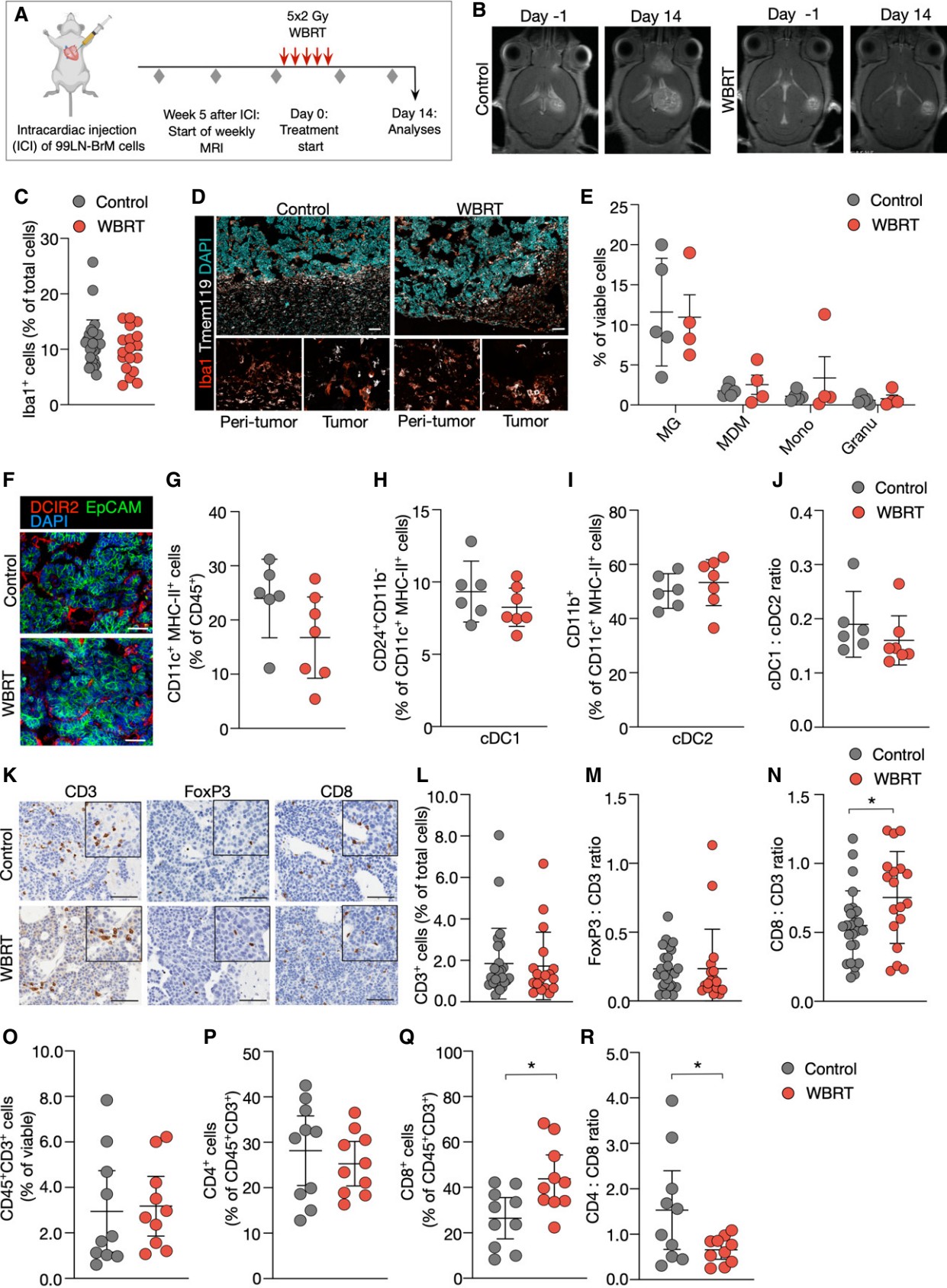

Figure 4.

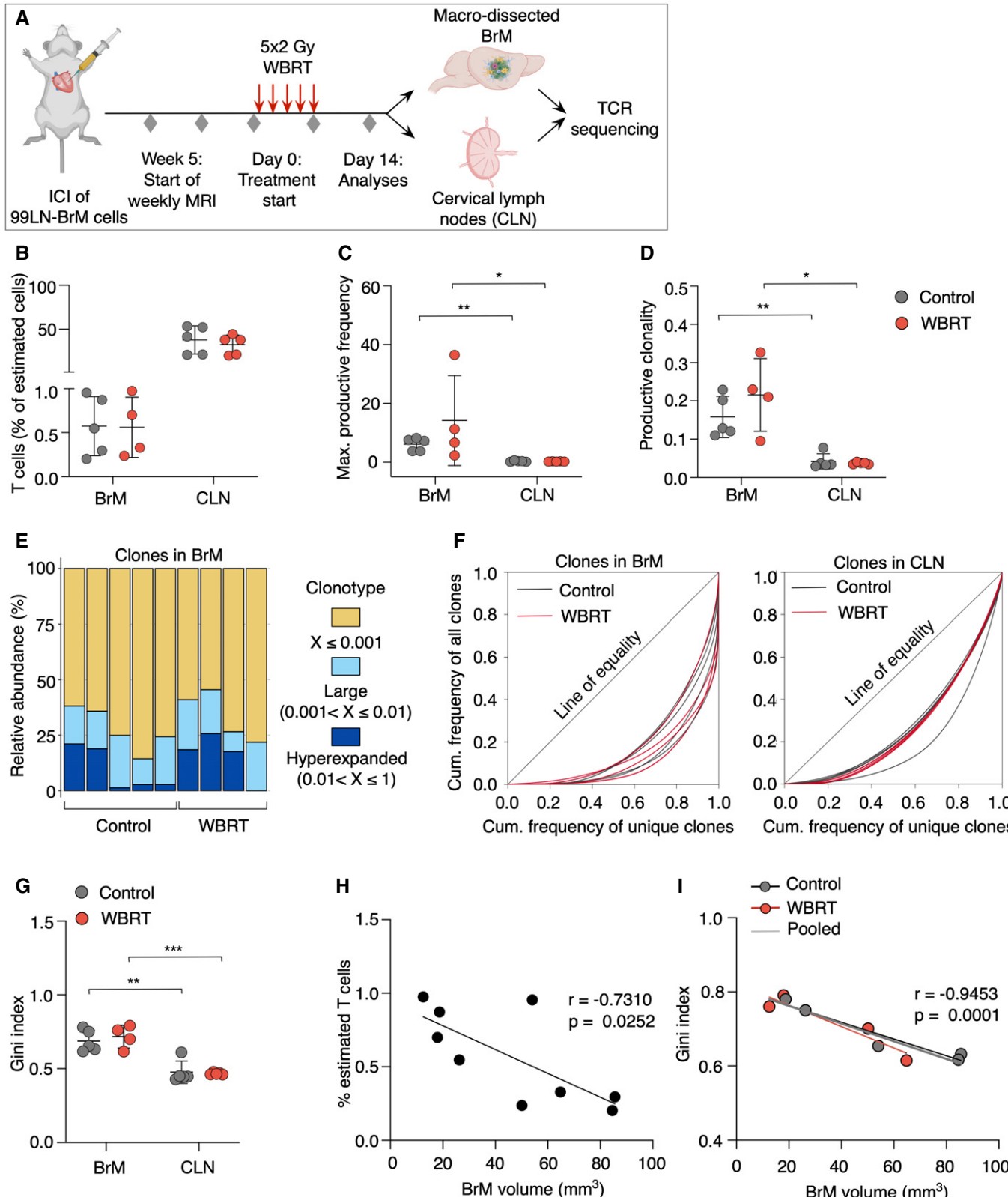

Figure 5.

**Figure 5. Analysis of T-cell receptor repertoires in BrM and CLN.**

A  Experimental design of TCR sequencing.
B  Percent T cells of estimated cells in BrM and CLN of mice from control or WBRT group.
C  Maximal productive frequency of clones in BrM and CLN samples.
D  Productive clonality in BrM and CLN samples.
E  Clonal space homeostasis of BrM samples indicating the relative abundance of clones with a specific frequency.
F  Lorenz curves generated with the reads of all clones in a sample to visualize deviation from perfect equality.
G  Lorenz curves were used to determine the Gini index of each sample, which is a measure of inequality reflecting clonal expansion.
H  Percent of estimated T cells correlated with BrM volume ($r = -0.7310$, $P = 0.0252$).
I  Gini indices of BrM samples correlated with BrM volume ($r = -0.9453$, $P = 0.0001$).

Data information: In (B–D + G), data are represented as scattered dot plot with line mean $\pm$ SD. *P*-values were obtained by Mann–Whitney test for (C, D) and unpaired *t*-test for (G) with **$P < 0.01$, and ***$P < 0.001$. For Control: $n = 5$ for BrM and CLN, for WBRT: $n = 4$ for BrM and $n = 5$ for CLN. In (H, I), the simple linear regression was plotted and Pearson's correlation was used to obtain correlation coefficient $r$ with *P*-value. Exact *P*-values can be found in Appendix Table S3.
Source data are available online for this figure.

indicating clonal expansion of TILs in BrM independent of variations in the number of sequenced TCRs (Fig 5F and G). Despite the indication of clonal expansion in BrM, we observed a negative correlation between BrM volume and percent of estimated T cells or Gini index (Fig 5H and I). Interestingly, clonal T-cell expansion was also observed in cervical lymph nodes (Fig 5F and G), however, to a lower extent compared with BrM (Fig 5G). Again, no differences were observed between control and irradiated samples (Fig 5G). Comparison of the top 100 TCR sequences revealed only few shared TCR sequences between matched BrM and CLN as well as control and irradiated BrM samples (Fig EV3D and E).

## Improved anti-tumor effects in response to radio-immunotherapy

We evaluated the efficacy of ICB using PD-1 neutralizing antibodies to reactivate T-cell responses in established BrM as monotherapy and in combination with radiotherapy. Mice received 2 Gy WBRT on five consecutive days, and treatment with ICB was started concurrently with the first radiation dose. Mice from the αPD-1 and αPD1 + WBRT group were treated with αPD-1 (250 µg/day) every third day throughout the trial period. Mice from the isotype and WBRT group received injections of the isotype control (IgG2a) (Fig 6A). All groups started with a similar average number of BrM lesions per animal and BrM volume on d-1 (Fig EV4A). Analysis of the percent increase in tumor burden revealed that WBRT significantly reduced BrM progression within the first 14 days after treatment, whereas αPD-1 monotreatment was inefficient to control

tumor growth (Figs 6B–D and EV4B). However, effects of WBRT were no longer evident after 28 days. In contrast, the combination of WBRT and PD-1 blockade led to a significant reduction in tumor outgrowth and improved anti-tumor efficacy of radio-immunotherapy was evident until 5–6 weeks after treatment start (Figs 6B–D and EV4B). Stratification of treatment responses into progressive disease (PD; change in BrM volume > +40%), stable disease (SD; change in BrM volume between −65 and +40%), partial response (PR; change in BrM volume > −65%), and complete response (CR; change in BrM volume = −100%) based on criteria previously proposed for a mouse brain tumor model (Aslan *et al*, 2020) revealed that all animals in the control group showed progressive disease. In the WBRT group, we found mice with SD and PR at early time points after treatment start. However, the percentage of mice with PD increased from d14 to d21 with all animals showing PD on d28. In the αPD-1 group, individual animals showed SD on d7 and d14. At later time points, PD was observed in all animals. In the combined modality group, we observed gradually improving response rates in individual mice, while other mice did not respond to the treatment (Fig 6E). Further analysis of the T-cell content in responders (R) and non-responders (NR) revealed that all responders in the combination group showed high T-cell infiltration, while the majority of the non-responders contained low T-cell counts. A similar correlation was not observed in mice that showed initial responses to WBRT (Fig 6F). Anti-tumor effects translated into significantly prolonged survival rates of the combination treatment group (Fig 6G). Mice that were categorized into the T-cell high subgroup showed prolonged overall symptom-free survival with

**Figure 6. Preclinical trial of the combination of WBRT and αPD-1 in 99LN-BrM.**

A  Experimental design of the combination trial.
B  Representative MRI images of 99LN-BrM from mice at different time points after treatment start.
C  Tumor growth curves of individual mice grouped by treatment. Maximal values of 8,000% increase are shown from treatment start to d63.
D  Quantification of relative tumor growth of the individual mice as area under the curve normalized to the survival time in weeks.
E  Stacked columns depict the percentage of animals with progressive disease (PD), stable disease (SD), partial response (PR), and complete response (CR) at the indicated time points.
F  Percentage of mice that were categorized as CD3 T-cell high or low stratified into responders and non-responders in the WBRT and WBRT + αPD-1 treatment group.
G  Kaplan–Meier curves show the percentage of symptom-free mice.

Data information: In (C–G), $n = 8$ for isotype, $n = 7$ for WBRT, $n = 8$ for αPD-1, $n = 9$ for WBRT + αPD-1 at treatment start. Numbers in the columns in (E) indicate the proportion of mice at each time point. Numerical data in (D) are represented as scattered dot plot with line mean $\pm$ SD; categorical data in (E + F) are represented as stacked columns. *P*-values were obtained by unpaired *t*-test in (D) and log-rank test in (G) with *$P < 0.05$, and **$P < 0.01$. Exact *P*-values can be found in Appendix Table S3.
Source data are available online for this figure.

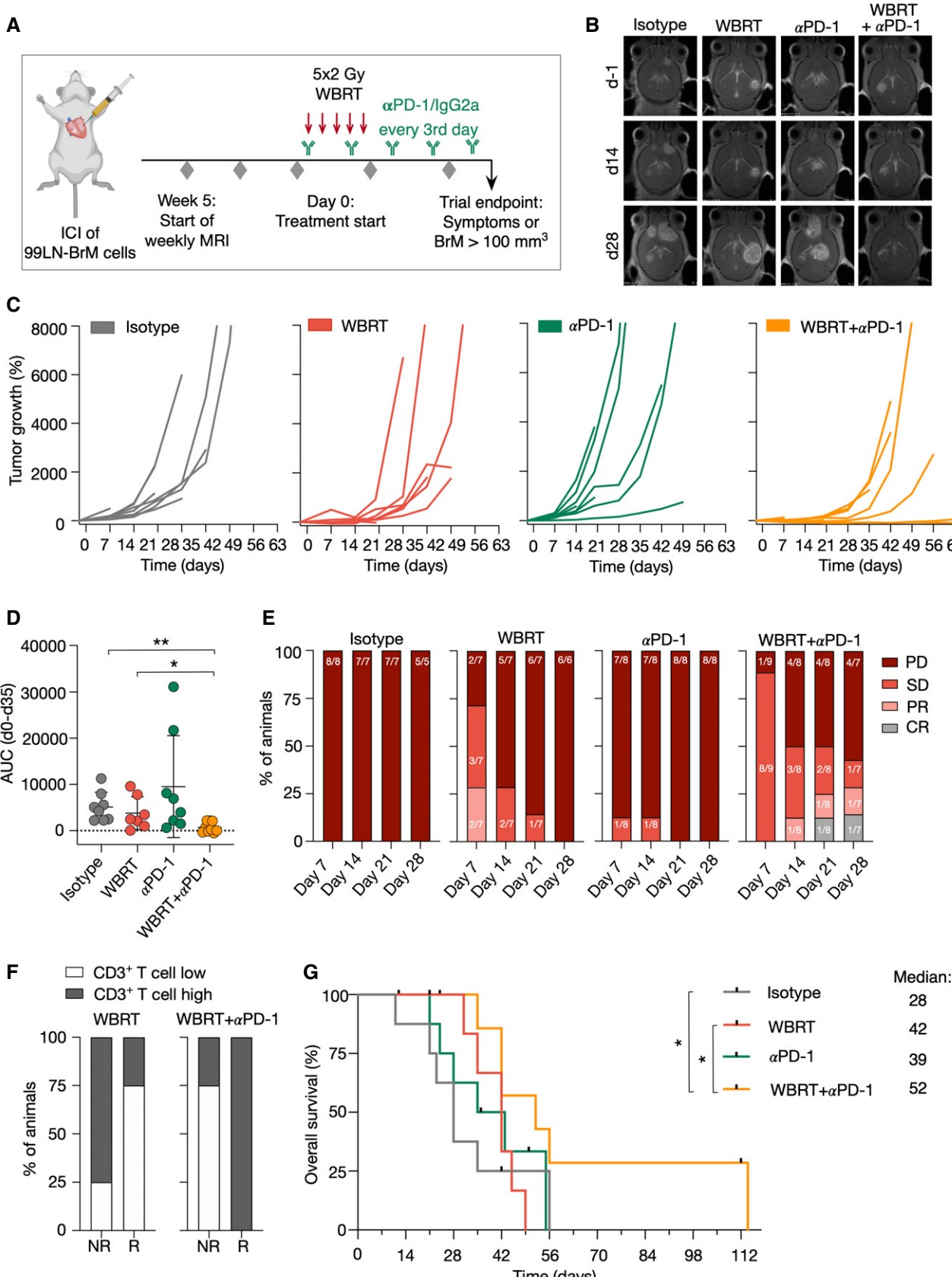

Figure 6.

some animals surviving up to 114 days after treatment start (Fig EV4C). Although mice in the αPD-1 monotherapy group showed median symptom-free survival of only 39 days, stratification in T-cell low and high subgroups revealed that mice with higher T-cell content showed significantly prolonged median survival of 42 days compared with 22.5 days for mice with low T-cell content (Fig EV4C).

### Immune checkpoint inhibition enhanced T-cell infiltration into BrM

We next evaluated effects of monotherapy compared with αPD-1 + WBRT on the immune cell composition of 99LN-BrM to identify mechanisms that underlie the improved efficacy of radio-immunotherapy. We did not observe differences in the spatial organization of T cells relative to TAMs and tumor cells across all experimental groups (Fig EV5A and B). However, flow cytometry revealed a mild increase in immune cell infiltration in all treatment groups compared with the control group with the most prominent effect being observed in response to combination therapy (Fig 7A). We observed increased numbers of T cells after checkpoint inhibition applied as monotherapy or in combination with WBRT, whereas the number of infiltrating DC did not change significantly (Fig 7B–D). Interestingly, WBRT monotherapy increased the proportion of CD8$^+$ T cells, while αPD-1 treatment alone rather increased the proportion of CD4$^+$ T cells. Combination of WBRT and αPD1 treatment resulted in an increase in CD4$^+$ and CD8$^+$ T cells compared with the control group (Fig 7D). αPD-1 treatment induced an increase in the amount of Foxp3$^+$ Tregs and led to a compensatory upregulation of PD-1 on T cells. Those effects were not observed in the WBRT and combination treatment groups (Fig 7E and F).

### Immune checkpoint inhibition leads to enhanced recruitment of PD-L1 expressing myeloid cells

To further dissect mechanisms that affect efficacy of radio-immunotherapy, we analyzed effects of myeloid cell-mediated immune suppression. Flow cytometry revealed minor changes in the infiltration rate of blood-borne myeloid cells in the different treatment groups compared

with isotype control (Fig 7G). The most prominent effect was observed in the percentage of PD-L1$^+$ myeloid cells (Fig 7H). We observed that monocytes and MDM represent the vast majority of PD-L1$^+$ myeloid cells in the TME. αPD-1 and combination treatment increased the amount of PD-L1$^+$ MDM compared with the control group (Fig 7H). As MDM represent the highest proportion of PD-L1$^+$ myeloid cells after αPD-1 and combination therapy, we tested the capability of primary bone marrow-derived macrophages (BMDM) to inhibit T cells in comparison with the microglial cell line EOC2 in an in vitro assay. Splenocyte-derived T cells were differentiated and activated in vitro and co-cultured with BMDM, EOC2, or 99LN-BrM tumor cells. T-cell activation was measured by flow cytometry of the activation marker CD69, IFNγ, and Gzmb after 24 h (Fig 7I). 99LN-BrM tumor cells alone did not significantly change T-cell activity based on CD69, IFNγ, and Gzmb protein level (Fig 7J and K). Across all analyzed conditions, BMDM showed the highest capacity to reduce CD69 and IFNγ on T cells compared with baseline T-cell activation. The most prominent reduction was observed in response to pre-stimulation with 99LN-BrM tumor cell-conditioned media (TuCM) and additional co-culture with tumor cells. EOC2 only reduced CD69 and IFNγ protein level on T cells in response to TuCM conditioning combined with tumor cell co-culture (Fig 7J and K). For Gzmb, a marginal reduction was observed after conditioning and in co-culture with 99LN-BrM cells with no significant differences between BMDM and EOC2. The addition of αPD-1 resulted in a slight increase in IFNγ and Gzmb in CD4$^+$ and CD8$^+$ cells and a significant increase in Gzmb levels on CD4$^+$ T cells. Interestingly, effects of PD-1 inhibition were only observed for BMDM after preconditioning with TuCM and co-culture with 99LN-BrM cells (Fig 7K), whereas the addition of αPD-1 to conditions with EOC2 cells did not lead to changes in IFNγ and Gzmb levels. This suggests that T-cell suppression by BMDM is at least in part mediated via the PD-1–PD-L1 axis, while T-cell suppression by EOC2 is likely mediated by alternative pathways.

## Discussion

Strategies that activate anti-tumor T-cell responses in brain cancers have been extensively tested in preclinical and clinical trials. Most

---

**Figure 7. Effects of radio-immunotherapy on T cells and myeloid cells in breast cancer-derived BrM.**

A  CD45$^+$ immune cells infiltrating 99LN-BrM ($n = 5$ for isotype, $n = 4$ for the other groups).

B  Flow cytometric analysis of dendritic cell (CD45$^+$CD11c$^+$CD83$^+$) infiltration ($n = 4$ for isotype, $n = 3$ for all other groups).

C  Quantitative analysis of IHC CD3$^+$ T cells at trial end point ($n = 10$ for isotype, $n = 7$ for WBRT, $n = 10$ for αPD-1, and $n = 8$ for WBRT + αPD-1 group).

D  Flow cytometric analysis of T-cell subpopulations in 99LN-BrM ($n = 4$ for isotype, $n = 3$ for all other groups).

E  Quantitative analysis of IHC FoxP3$^+$ T cells at trial end point ($n = 10$ for isotype, $n = 7$ for WBRT, $n = 10$ for αPD-1, and $n = 8$ for WBRT + αPD-1 group).

F  Flow cytometric analysis of PD-1 expression on T cells in response to different treatments (control $n = 4$, all other groups $n = 3$).

G  Composition of the myeloid compartment in 99LN-BrM in the different treatment groups ($n = 5$ for isotype, all other groups $n = 4$).

H  Relative abundance (%) of PD-L1$^+$ myeloid cell types in BrM samples of the four treatment groups ($n = 5$ for isotype, all other groups $n = 4$).

I  Experimental design of the in vitro T-cell activation assay.

J  Relative CD69 protein level on T cells cultivated with different cell types, unstimulated, or stimulated with tumor-conditioned media (Cond.) ($n = 3$).

K  Relative Gzmb and IFNγ protein level on CD4$^+$ and CD8$^+$ T cells cultivated with different cell types, with or without αPD-1 and unstimulated or stimulated with tumor-conditioned media (Cond.) ($n = 4$ for conditions including BMDM, $n = 5$ for all other conditions).

Data information: Data in (A–C, E + F) are represented as scattered dot plot with line at mean ± SD. Data in (D + G + H) are represented as stacked columns ± SD. Data in (J + K) are represented as scattered dot plot with line at median. Dotted line depicts baseline T-cell activation. P-values were obtained by Mann–Whitney test in (C) unpaired t-test in (F), two-way ANOVA in (D, G and H), and paired t-test in (J and K) with *P < 0.05 and **P < 0.01, and ***P < 0.001. Exact P-values can be found in Appendix Table S3.

Source data are available online for this figure.

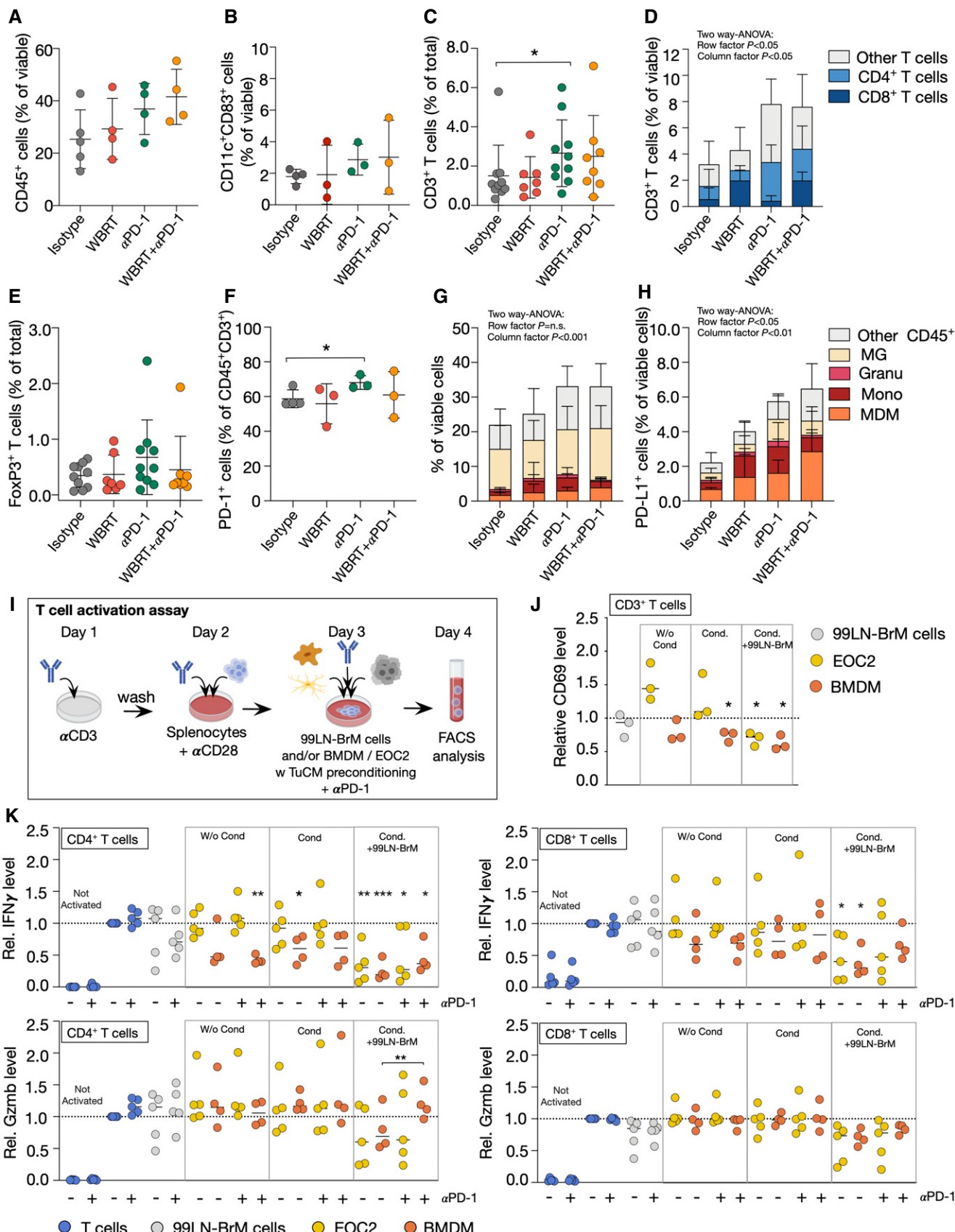

Figure 7.

approaches aim to modulate the lymphoid compartment, e.g., by introducing engineered T cells with chimeric antigen receptors (CAR) or by blocking immune checkpoints on T cells (Khalil et al, 2016; Waldman et al, 2020). However, in the context of the highly immune suppressive microenvironment of brain tumors, it is expected that effective reactivation of anti-tumor T-cell responses can only be achieved in combination with strategies that revoke the immune suppressive pressure within the tumor microenvironment. Several cell types have been associated with immune suppression in brain tumors including tumor cells, brain-resident glial cells such as astrocytes and microglia (Priego et al, 2018; Heiland et al, 2019; Guldner et al, 2020) as well as recruited myeloid cell populations (Bowman et al, 2016; Klemm et al, 2020; Schulz et al, 2020). Here, we performed preclinical testing of the efficacy of ionizing radiation to sensitize breast-to-brain metastases to immunotherapy. Our work revealed that WBRT combined with ICB leads to immune modulation of lymphoid and myeloid populations and results in improved tumor control and prolonged median survival.

Efficient tumor control of glioma and BrM was shown to depend on enhanced CD8$^+$ T-cell priming and recruitment to CNS lesions which synergized with ICB (Taggart et al, 2018; Song et al, 2020). In line with previous findings, our data indicate that response rates to ICB correlate with the amount of infiltrating T cells. Stimuli underlying enhanced T-cell priming and trafficking to CNS tumors can be diverse and include the presence of an extracranial primary tumor (Taggart et al, 2018) or VEGF-C induced modulation of meningeal lymphatic vasculature (Song et al, 2020). Here, we show that WBRT and ICB induce different alterations in the immune composition of BrM. However, only the combination of both treatment modalities mounts a sufficient immune response to control disease progression and improve median survival rates in a breast-to-brain metastasis model. Immune modulation in the context of radio-immunotherapy was characterized by increased infiltration of CD4$^+$ and CD8$^+$ T cells but prevented the induction of lymphoid-mediated immune suppressive mechanisms such as infiltration of Foxp3$^+$ Tregs or compensatory PD-1 expression on T cells. Thus, our data support the observation that both CD4$^+$ and CD8$^+$ T cells are critical for efficient anti-tumor immunity and are implicated in driving responses to immunotherapy (Ostroumov et al, 2018; Aslan et al, 2020). Although radio-immunotherapy showed improved anti-tumor effects, we observed tumor regrowth after an initial period of tumor stasis. Our data suggest that T-cell reactivation leads to induction of compensatory immune suppressive mechanisms that ultimately blunt the efficacy of radio-immunotherapy. There is growing evidence that brain-resident and recruited myeloid cells in brain cancers show transcriptional dichotomy with TAM-MG rather being associated with pro-inflammatory signatures while TAM-MDM are implicated in immune suppression, antigen presentation, and wound-healing responses in glioma and BrM (Bowman et al, 2016; Klemm et al, 2020; Schulz et al, 2020). We observed that acquired resistance to radio-immunotherapy is at least in part mediated by PD-L1 expressing blood-borne monocytes and MDM that infiltrate BrM lesions. This was further supported by transcriptomic analyses that indicate a crucial role for MDM in regulating T-cell activity. We previously demonstrated that WBRT revokes MDM-mediated immune suppression by inducing the recruitment of tumor-naïve blood-borne myeloid cells in a lung-to-brain metastasis model (Schulz et al, 2020). However, this effect was only transient due to the rapid

induction of tumor education gene signatures in TAMs upon recruitment to BrM (Schulz et al, 2020). Consequently, the development of therapeutic strategies that allow for targeted modulation of myeloid-mediated immune suppression could be key to achieve more sustained anti-tumor immune responses. Indeed, Aslan et al (2020) previously tested the efficacy of colony-stimulating factor 1 receptor (CSF1R) blockade using a neutralizing antibody to target TAMs and demonstrated improved treatment responses to ICB in a glioma model. While general TAM-targeting therapies could provide beneficial effects on treatment response, our data rather suggest the necessity of strategies that allow to selectively modulate immune suppressive TAM functions while sparing physiological functions, i.e., of brain-resident MG that are critical for CNS homeostasis. Transcriptional analyses of TAMs and TILs provide a framework of potential myeloid and lymphoid checkpoints that could be considered for combination therapies to achieve a more efficient response, as previously shown for combined PD-1 and CTLA-4 blockade in preclinical studies (Taggart et al, 2018) and clinical trials (Long et al, 2018; Tawbi et al, 2018). In addition to myeloid cell-mediated effects, immune suppression by other cells within the brain tumor microenvironment should also be taken into account as previously demonstrated for astrocyte subsets (Priego et al, 2018; Heiland et al, 2019). Given recent clinical data, comparison of WBRT and stereotactic radiosurgery will be of critical importance to define optimal treatment modalities to induce immune modulatory effects that synergize with ICB in brain metastases (Lehrer et al, 2018; Kotecha et al, 2019).

## Materials and Methods

### Mice

All animal studies were approved by the government committee (Regierungspräsidium Darmstadt, Germany; protocol numbers F123/1016 and F123/1068) and were conducted in accordance with the requirements of the German Animal Welfare Act. C57Bl6/J and FVB/n mice were purchased from Charles River Laboratories. 10- to 12-week-old female C57Bl6J and 8-week-old female FVBn mice were used. Mice were kept in individually ventilated cages under specific pathogen-free conditions at room temperature of 22°C ± 2°C, 45–65% relative humidity, and a light/dark cycle (12 h of light/12 h of dark). Food and drinking water were provided ad libitum. The microbiological status was periodically monitored by sentinels.

### Primary cell cultures

Bone marrow-derived macrophages (BMDM) were differentiated from monocytes isolated from femurs of 6- to 8-week-old mice (C57Bl6/J). Cells were cultured 7 days in DMEM + 10% FBS, 1% L-glutamine, and 1% penicillin/streptomycin, supplemented with 10 ng/ml CSF-1. T cells for the in vitro activation assay were differentiated from splenocytes isolated from 6- to 8-week-old C57Bl6/J mice. After erythrocyte lysis, $2 \times 10^5$ splenocytes per well were seeded on αCD3-coated 96-well plates (24 h before, 0.5 µg/ml, clone: 145-2C11, Thermo Fisher) in RPMI + 10% FBS, 1% L-glutamine, 1% penicillin/streptomycin, 0.1% β-mercaptoethanol, and 2 µg/ml αCD28 (clone:37.51, Thermo Fisher).

### Cell lines

The 99LN cell line was derived from a metastatic lymph node of the MMTV-PyMT breast cancer model (C57Bl6/J background) and selected *in vivo* for brain homing capacity as previously described (Bowman *et al*, 2016), resulting in the 99LN-BrM2 variant used herein (Chae *et al*, 2019). The TS1 cell line was derived from primary tumors of the MMTV-PyMT breast cancer model (FVB/n background) as previously described (Shree *et al*, 2011), and brain metastatic variants were selected as previously described (Sevenich *et al*, 2014). The 99LN-BrM and TS1-BrM cell lines were maintained in DMEM containing 10% fetal bovine serum with 1% L-glutamine and 1% penicillin/streptomycin. The EOC2 microglial cell line was purchased from ATCC. The cells were maintained in DMEM with 10% FBS, 2% glutamine, and 1% penicillin/streptomycin supplemented with 20 ng/ml IL-34 and 5 ng/ml TGF-β. All cell lines were regularly tested for possible mycoplasma contamination. All cell lines used in this study were mycoplasma negative.

### Generation of experimental brain metastasis and *in vivo* MRI measurements

For all 99LN-BrM or TS1-BrM *in vivo* experiments, 10- to 12-week-old C57Bl6/J mice or 8-week-old FVB/n mice were used. To generate BrM, $6 \times 10^4$ or $1 \times 10^5$ 99LN-BrM cells and $3 \times 10^4$ TS1-BrM cells were injected into the left ventricle of mice (i.c. injection), as previously described (Sevenich *et al*, 2014; Bowman *et al*, 2016). BrM progression was monitored by MRI measurements until the trial end point. MRI was performed using a 7 Tesla Small Animal MR Scanner (PharmaScan, Bruker) with a volume coil as transmitter and a head surface coil for signal reception as previously described (Chae *et al*, 2019). Mice were injected intraperitoneally (i.p.) with 100 μl Gadobutrol (Gadovist, 1 mmol/ml, Bayer) before the measurement. Data acquisition was performed using the Paravision 6.0.1 software with images being acquired in coronal planes. A T1-weighted RARE sequence (T1 RARE; TE/TR = 6.5 ms/1,500 ms) was applied for obtaining T1-weighted images. Volumetric analysis of BrM was performed on MR image DICOM files using a segmentation tool in the ITK-Snap software (Yushkevich *et al*, 2019). For detection of extracranial metastases, an abdominal volume coil without injection of contrast agent was used. A T1 Flash image with relaxation enhancement (T1 Flash; TE/TR = 1.8 ms/187.766 ms) was applied.

### Preclinical trials

#### Whole brain radiotherapy

WBRT was applied with the Small Animal Radiation Research Platform (SARRP, X-Strahl Ltd, Camberley, UK) (Wong *et al*, 2008) as previously described (Chae *et al*, 2019). The SARRP is equipped with an on-board Cone Beam CT (CBCT) system for diagnostic imaging and radiation treatment planning. The integrated Muriplan software allows contouring, image-guided treatment design, dose calculation, and application of highly focused radiation fields. Mice were anesthetized with isoflurane (2.5%) and imaged by performing a CBCT operating at 60 kV and 0.8 mA. CBCT images were transferred to the Muriplan software, and individual isocenters were selected for radiotherapy. Radiation was applied as fractionated

WBRT with 2 Gy on five consecutive days and a $10 \times 10$ mm collimator as 1 arc operating at 220 kV and 13 mA with 5.2 cGy/s.

#### Combination trial

Mice were stratified into isotype, WBRT, αPD-1, and WBRT + αPD-1 treatment groups starting with a similar tumor volume, based on volumetric MRI measurements at d-1. Treatment was commenced at d0. WBRT was applied as described above. For ICB, mice were treated with 250 μg of αPD-1 (RMPI1-14, BioXCell) every third day, starting with the first radiation dose on d0. Animals in the isotype and WBRT groups were treated with 250 μg IgG2a (2A3, BioXCell). For the WBRT + αPD-1 treatment group, the first αPD-1 dose was applied concurrent with the first WBRT dose. For the short-term trials, mice were sacrificed on d14 after treatment initiation. For the survival trials, animals were treated until they developed symptoms from BrM or reached a maximum volume of > 100 $mm^3$ based on MRI measurements.

#### T-cell depletion

Mice were stratified into two groups 7 days after intracardiac injection of $1 \times 10^5$ 99LN-BrM cells and treated with 150 μg αCD4 (GK1.5, BioXCell) and 150 μg αCD8 (YTS 169.4, BioXCell) or with 300 μg of IgG2b (LTF-2, BioXCell). The first three injections were applied on consecutive days followed by weekly injections. T-cell depletion was confirmed by flow cytometry of peripheral blood. BrM onset and progression were monitored by weekly MRI measurements.

### Flow cytometry and sorting

For blood analysis, mice were bled via submandibular routes and blood was transferred into EDTA-coated tubes. Blood was filtered through 70-μm cell strainers followed by red blood cell (RBC) lysis. Cells were incubated for 15 min with Fc-block at 4°C. Afterward, cells were incubated 20 min at 4°C with directly labeled antibodies and live–dead staining. For flow cytometry of BrM, mice were anesthetized with ketamine/xylazine and perfused with PBS, and BrM were macrodissected. Tissue was dissociated with the Brain Tumor Dissociation Kit (Miltenyi) followed by filtering and RBC lysis as described for blood. Additionally, samples were incubated 15 min at 4°C with myelin removal beads II (Miltenyi). Single cell suspensions were incubated with Fc-block and antibodies. For analysis of *in vitro* experiments, cells were incubated with Fc-block and followed by antibody staining. All flow cytometry analyses were performed on a BD Fortessa using the HTS-unit. Compensation was done with ArC and AbC compensation bead kits (Thermo Fisher). Data were acquired in BD FACS Diva and analyzed with FlowJo (BD Life Sciences v10.6.2). For FACS sorting, samples were processed as described above. FACS sorting was performed on a BD FACS Aria Fusion. Cells were sorted directly in TRIzol LS and snap frozen on dry ice. Antibodies and live–dead staining from all flow panels can be found in Appendix Table S1.

### RNA sequencing

RNA was isolated by chloroform extraction and isopropanol precipitation. RNA sequencing libraries were generated with the

SMART-Seq preparation kit (CloneTech) and fragmented with the Nextera XT kit (Illumina). Paired-end, 150 base pair, sequencing was performed by Genewiz (New Jersey, USA) on an Illumina HiSeq4000. Downstream data processing was conducted using the HUSAR platform (DKFZ, Heidelberg, Germany). Reads were mapped to the mouse genome (# 38) using TopHat2 (v. 2.0.14). Transcript abundance was quantified using genecode annotations (release vM23) with HTSeq-Count, based on HTSeq (Anders *et al,* 2015). Differential gene expression was assessed with DESeq2 (Love *et al,* 2014). Vst data were used for PCA clustering, heatmaps, and graphs representing relative expression values. For pathway analyses and gene annotation, data were manually filtered for top genes, respecting a basemean greater than 20 and an adjusted *P*-value (*P*adj) less than 0.05 (= FDR 5%). Pathway analysis and gene annotation were performed with Metascape (Zhou *et al,* 2019) or clusterprofiler (Yu *et al,* 2012) and enrichplot (Yu, 2019) in R (version 3.4.3). Euler plots were generated with eulerR (Larsson, 2020).

### TCRβ profiling and analysis

For TCR profiling, mice were treated with WBRT or left untreated. BrM and CLN were isolated 14 days after treatment start and DNA was isolated with the DNeasy Blood & Tissue Kit (Qiagen). TCRβ sequencing (ImmunoSEQ) was performed at Adaptive Biotechnologies. Analysis was performed with the ImmunoSEQ Analyzer 3.0. For the analysis of clone size distribution and for the generation of Lorenz curves, R packages were applied (R studio version 3.6.2, immunarch version 0.5.5, ineq version 0.2.13).

### Tissue preparation and immunostaining

Tissue for frozen histology was fixed in 4% PFA overnight and subsequently transferred into 30% sucrose until the tissue was fully equilibrated. Tissues were then embedded in OCT (Tissue-Tek), and 10-µm cryostat tissue sections were used for subsequent analyses. For immunofluorescence staining, frozen sections were thawed, dried at room temperature (RT), and rehydrated. For standard staining protocols, tissue sections were blocked in 3% BSA + 0.1% Triton X-100 in PBS for 1 h at room temperature, followed by incubation with primary antibodies in 1.5% BSA overnight at 4°C. Primary antibody information is listed in Appendix Table S2. Fluorophore-conjugated secondary antibodies were used at a dilution of 1:500 in 1.5% BSA in PBS for 1 h at RT.

Formalin-fixed and paraffin-embedded (FFPE) sections were processed for immunohistochemistry (IHC) using a Leica Bond Max automated staining device. The automated deparaffinization/rehydration, citrate buffer-based antigen retrieval, and blocking of unspecific protein binding and endogenous peroxidase were followed by incubation with primary antibodies (Appendix Table S2), followed by HRP-labeled secondary antibodies and DAB conversion. Hematoxylin and eosin (H&E) staining was performed on an automated staining device (Leica Autostainer XL).

For multiplexed histology, FFPE sections were stained using the Opal Polaris 7 Color Kit (Akoya Biosciences Inc.) with 7-plex stainings being performed on a LabSatTM Research Automated staining instrument (Lunaphore Technologies SA). Primary antibody information is listed in Appendix Table S2.

### Microscopy and image analysis

Tissue sections were visualized using the confocal quantitative image cytometer CQ1 (Yokogawa) or the Aperio Scan Scope. Quantification of Iba1$^+$ macrophages/MG and T-cell populations was performed with Aperio ImageScope (v12.4.0.5043) using either a nuclear counting algorithm or a pixel counting algorithm depending on the complexity of cell shape. Multiplex stainings were acquired on Vectra Polaris (Akoya Biosciences Inc.) using Motif technology. Multispectral image analysis was performed with Phenochart, InForm Image analysis software (Akoya Biosciences Inc.), and HALO software (Indica labs).

### RNA isolation, cDNA synthesis, and quantitative real-time PCR

RNA was isolated with TRIzol, DNase treated, and 0.5–1.5 µg of RNA was used for cDNA synthesis using the High-Capacity cDNA Reverse Transcription Kit (Thermo Fisher Scientific). The following TaqMan assays were used for qRT-PCR: PD-1 (Mm01285676_m1) and PD-L1 (Mm03048248_m1). Assays were run in triplicate, and expression was normalized to ubiquitin C (*Ubc* Mm02525934_g1) and glyceraldehyde 3-phosphate dehydrogenase (*Gapdh* Mm99999915_g1) for each sample.

**The paper explained**

**Problem**
Response rates to checkpoint inhibitors for poorly immunogenic cancers are low. Additionally, cancer cells metastasizing to the brain are protected from the adaptive immune system by the immune suppressive tumor microenvironment (TME). Therefore, many brain metastasis (BrM) patients cannot benefit from the advent of immunotherapies and are restricted to standard of care therapies with limited success rates.

**Results**
We found that cell types essential for successful immunotherapy via PD-1 blockade are present in murine breast cancer-derived BrM, including T cells and dendritic cells. T cells in the TME of a breast cancer BrM mouse model clonally expanded, indicating prior T-cell activation. A high proportion of T cells expressed PD-1, whereas tumor and myeloid cells recruited to the lesions expressed PD-L1. Classical fractionated whole brain radiotherapy (WBRT) increased the proportion of cytotoxic CD8$^+$ T cells and prolonged survival transiently. Combination therapy of WBRT and PD-1 blockade increased T-cell infiltration and prevented the induction of compensatory inhibitory responses in lymphocytes induced by anti-PD-1 monotherapy. This led to reduced tumor progression and prolonged survival compared with WBRT alone. Analysis of combination-treated BrM revealed increased infiltration with PD-L1-positive myeloid cells recruited from the periphery.

**Impact**
We demonstrate that radiotherapy can sensitize low immunogenic BrM to checkpoint blockade. We showed that myeloid cells recruited from the periphery play a crucial role in regulating T-cell activation and are implicated in generating an immune suppressive environment, underlying the importance of investigating the role of these cells in acquired resistance to checkpoint blockade in more details.

### *In vitro* T-cell activation

T cells were differentiated and activated from spleens of C57BL6/J mice. Before organ extraction, mice were perfused with PBS. RBC lysis and splenocyte activation were performed as described above and modified from BestProtocols® (Thermo Fisher) using stimulation with αCD3 and αCD28. As control, splenocytes were not activated with αCD3 and αCD28 and denoted as "not activated". After 24 h of differentiation and activation, $5 \times 10^4$ 99LN-BrM, EOC2, or BMDM cells/well were added to the culture in the same media as for T-cell activation but supplemented with 10 ng/ml CSF-1 for BMDM and EOC2 survival. To test the influence of preconditioned BMDM and EOC2 on T cells, these cells were cultured in tumor-conditioned media (TuCM) overnight before addition to the assay. TuCM was generated by seeding $5 \times 10^6$ 99LN cells in 8 ml DMEM media supplemented with 10% FBS, 1 or 2% L-glutamine, 1% penicillin/streptomycin overnight, followed by sterile filtration. In conditions with tumor cell co-culture, pre-stimulated BMDM or EOC2 and T cells were co-cultured with 99LN-BrM tumor cells. To test effects of PD-1 inhibition *in vitro*, αPD-1 (400 ng/ml) was added. Cells were harvested after 24 h. T cells were identified as $CD45^+CD11b^-$ cells or $CD45^+CD4^+$ and $CD45^+CD8^+$ cells. Activation was analyzed by flow cytometry based on CD69, IFNγ, or Gzmb protein level.

### Data presentation and statistical analysis

Data are represented as mean ± SD or as indicated in the figure legend. Numerical data were analyzed using the statistical tests noted within the corresponding sections of the manuscript. Statistical analyses were performed with GraphPad Prism software v8 or R (version 3.4.3) performing tests as indicated and were considered statistically significant, with $*P < 0.05$, $**P < 0.01$ and $***P < 0.001$.

## Data availability

The datasets generated in this study are available in the following databases:

- RNAseq: Gene Expression omnibus: GSE164049 (https://www.ncbi.nlm.nih.gov/geo/query/acc.cgi?acc=GSE164049)
- TCRseq: https://doi.org/10.21417/KN2021EMBOMM (https://clients.adaptivebiotech.com/pub/niesel-2021-embomm).

**Expanded View** for this article is available online.

### Acknowledgements

We thank Petra Dinse, Annette Trzmiel, Julius Oppermann, Stephanie Hehlgans, Jeannie Peifer, and Judith Bergs for excellent technical support and members of the Sevenich lab and the Georg-Speyer-Haus for insightful discussion. We thank Johanna Joyce for providing the 99LN-BrM and TS1 cell lines. Schematics in the figures were generated with BioRender. Research in the lab of LS is supported by institutional funds from the Georg-Speyer-Haus jointly funded by the German Federal Ministry of Health and the Ministry of Higher Education, Research and the Arts of the State of Hesse (HMWK), as well as grants from the LOEWE Center Frankfurt Cancer Institute (FCI), the German Cancer Aid (Max-Eder Junior Group Leader Program 70111752), German Research Foundation (SE2234/3-1), the Beug Foundation for Metastasis Research, and the Dr. Bodo Sponholz Foundation. Open Access funding enabled and organized by Projekt DEAL.

## Author contributions

KN, MS, TA, JA, AS-B, and AM performed *in vitro* and *in vivo* experiments and analyzed data; MS analyzed RNAseq data; KN and TO analyzed TCRseq data. KN, TA, and ML performed MRI measurements; SS assisted in the flow cytometric analysis. FR assisted and advised on application of radiotherapy; JM, KHP, and YR performed, analyzed or advised on Phenoptics analysis. KN and LS wrote the manuscript. LS conceived and supervised the project. All authors edited and commented on the manuscript.

## Conflict of interest

The authors declare that they have no conflict of interest.

## For more information

Sevenich lab website: https://georg-speyer-haus.de/en/staff/sevenich-research/

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
