## [Review Process File · EMBO Molecular Medicine]

The immune suppressive microenvironment affects efficacy of radio-immunotherapy in brain metastasis

Lisa Sevenich, Katja Niesel, Michael Schulz, Julian Anthes, Tijna Alekseeva, Jadranka Macas, Anna Salamero-Boix, Aylin Moeckl, Timm Oberwahrenbrock, Marco Lories, Stefan Stein, Karl Plate, Yvonne Reiss, and Franz Rödel

DOI: [10.15252/emmm.202013412](https://doi.org/10.15252/emmm.202013412)

Corresponding author: Lisa Sevenich (sevenich@gsh.uni-frankfurt.de)

Review Timeline:

Submission Date:	7th Sep 20
Editorial Decision:	7th Oct 20
Revision Received:	19th Jan 21
Editorial Decision:	11th Feb 21
Revision Received:	18th Feb 21
Accepted:	22nd Feb 21

Editor: Lise Roth

Transaction Report:

7th Oct 2020

Dear Dr. Sevenich,

Thank you for the submission of your manuscript to EMBO Molecular Medicine. We have now received feedback from the three reviewers who agreed to evaluate your manuscript. As you will see from the reports below, the referees acknowledge the interest of the study and are overall supporting publication of your work pending appropriate revisions.

Addressing the reviewers' concerns in full will be necessary for further considering the manuscript in our journal, and acceptance of the manuscript will entail a second round of review. Particular attention should be given to softening the language and to avoiding overstatements. EMBO Molecular Medicine encourages a single round of revision only and therefore, acceptance or rejection of the manuscript will depend on the completeness of your responses included in the next, final version of the manuscript. For this reason, and to save you from any frustrations in the end, I would strongly advise against returning an incomplete revision.

When submitting your revised manuscript, please carefully review the instructions that follow below. Failure to include requested items will delay the evaluation of your revision:

- 1) A .docx formatted version of the manuscript text (including legends for main figures, EV figures and tables). Please make sure that the changes are highlighted to be clearly visible.
- 2) Individual production quality figure files as .eps, .tif, .jpg (one file per figure).
- 3) A .docx formatted letter INCLUDING the reviewers' reports and your detailed point-by-point responses to their comments. As part of the EMBO Press transparent editorial process, the point-by-point response is part of the Review Process File (RPF), which will be published alongside your paper.
- 4) A complete author checklist, which you can download from our author guidelines (<https://www.embopress.org/page/journal/17574684/authorguide#submissionofrevisions>). Please insert information in the checklist that is also reflected in the manuscript. The completed author checklist will also be part of the RPF.
- 5) Before submitting your revision, primary datasets produced in this study need to be deposited in an appropriate public database (see <https://www.embopress.org/page/journal/17574684/authorguide#dataavailability>). Please remember to provide a reviewer password if the datasets are not yet public. The accession numbers and database should be listed in a formal "Data Availability" section (placed after Materials & Method). Please note that the Data Availability Section is restricted to new primary data that are part of this study.

6) We would also encourage you to include the source data for figure panels that show essential data. Numerical data should be provided as individual .xls or .csv files (including a tab describing the data). For blots or microscopy, uncropped images should be submitted (using a zip archive if multiple images need to be supplied for one panel). Additional information on source data and instruction on how to label the files are available at .

7) Our journal encourages inclusion of *data citations in the reference list* to directly cite datasets that were re-used and obtained from public databases. Data citations in the article text are distinct from normal bibliographical citations and should directly link to the database records from which the data can be accessed. In the main text, data citations are formatted as follows: "Data ref: Smith et al, 2001" or "Data ref: NCBI Sequence Read Archive PRJNA342805, 2017". In the Reference list, data citations must be labeled with "[DATASET]". A data reference must provide the database name, accession number/identifiers and a resolvable link to the landing page from which the data can be accessed at the end of the reference. Further instructions are available at .

8) We replaced Supplementary Information with Expanded View (EV) Figures and Tables that are collapsible/expandable online. A maximum of 5 EV Figures can be typeset. EV Figures should be cited as 'Figure EV1, Figure EV2" etc... in the text and their respective legends should be included in the main text after the legends of regular figures.

- Additional Tables/Datasets should be labeled and referred to as Table EV1, Dataset EV1, etc. Legends have to be provided in a separate tab in case of .xls files. Alternatively, the legend can be supplied as a separate text file (README) and zipped together with the Table/Dataset file. See detailed instructions here:

9) For more information: There is space at the end of each article to list relevant web links for further consultation by our readers. Could you identify some relevant ones and provide such information as well? Some examples are patient associations, relevant databases, OMIM/proteins/genes links, author's websites, etc...

10) Every published paper now includes a 'Synopsis' to further enhance discoverability. Synopses are displayed on the journal webpage and are freely accessible to all readers. They include a short stand first (maximum of 300 characters, including space) as well as 2-5 one-sentences bullet points that summarizes the paper. Please write the bullet points to summarize the key NEW findings. They should be designed to be complementary to the abstract - i.e. not repeat the same text. We encourage inclusion of key acronyms and quantitative information (maximum of 30 words / bullet point). Please use the passive voice. Please attach these in a separate file or send them by email, we will incorporate them accordingly.

Please also suggest a striking image or visual abstract to illustrate your article. If you do please provide a png file 550 px-wide x 400-px high.

11) As part of the EMBO Publications transparent editorial process initiative (see our Editorial at <http://embomolmed.embopress.org/content/2/9/329>), EMBO Molecular Medicine will publish online a Review Process File (RPF) to accompany accepted manuscripts.

In the event of acceptance, this file will be published in conjunction with your paper and will include the anonymous referee reports, your point-by-point response and all pertinent correspondence relating to the manuscript. Let us know whether you agree with the publication of the RPF and as here, if you want to remove or not any figures from it prior to publication.

I look forward to receiving your revised manuscript.

Yours sincerely,

Lise Roth

Lise Roth, PhD
Editor
EMBO Molecular Medicine

Referee #1 (Comments on Novelty/Model System for Author):

This is a technically very complex investigation with many valuable results. In particular, the experiments underline the importance of the myeloid compartment in brain metastasis. In addition, the data underline clinical observations and provide some biological explanations. In particular, the very elaborate models, the imaging, the irradiation and the technical processing are very impressive. The weakness of the manuscript is more the interpretation of the results and the discussion of the limitations. The intracranial injection model has a very high biological variance and, in particular, extracranial micro-metastasis formations cannot be excluded. For example, no reference is made to this point. But whether extracranial micrometastases are present could be of enormous importance for the immune reaction (ICB). There are further important limitations of this study (see text below). From my point of view, the limitations at least should be discussed. In view of this, the discussion of the possible signaling pathways of the myeloid signaling pathway should be more pronounced and the role the glia, e.g. Astrocytes almost missing. Astrocytes are also important candidates for the immunosuppressive reaction of the brain tissue. For these reasons, I recommend at least a restructuring of the discussion and abstract, especially synergism must be discussed (see below). Another important point is also not mentioned. Usually solitary metastases are treated with SRS and not WBRT. SRS and WBRT definitely have different biological and clinical effects. This should at least be mentioned. Other parts of the discussion can be significantly shortened.

Point 1: Measurement of extracranial tumor volume is missing and synergistic effect

The introduction cited the very important findings of Taggart et al. 2018. These findings underlined the impact of the extracranial Tumor for effective intracranial ICB. Taggart et al. demonstrated that without extracranial tumor tissue, there will be no intracranial response of ICB. Here, all the data derive from an intracardiac injection of the 99LN BrM metastasis model. Despite the 99LN BrM brain seeking cell line used for the experiments, it is unlikely that there is no extracranial metastasis growth at all (e.g. micro-metastasis in the bone marrow). The current manuscript do not report, how the extracranial metastasis load is measured. Also both citations (Bowman et al 2016 and Chae et al 2019) did not describe the frequency of extracranial (micro)-metastasis formation of the 99LN BrM model. However, to interpret all the immunological-findings (ICB) in respect to Taggart et al, it would be important to know, the frequency, and volume of the extracranial tumor mass. In particular, it could be one important difference between non-responder and responder to ICB (Figure 6). This is also important for the statement that radiotherapy is really synergistic. If you only add the differences between the OS data:

WBRT 42 days - Iso-Type control 28 days =difference of 14 days

aPD1 39 days - Iso-Type control 28 days =difference of 11 days

An additive effect would be around 28 days (control) + 14 days (difference WBRT) + 11 days (difference ICB) = 53 days

Interestingly the Median OS of WBRT + anti PD1 is 52 days. Thus, it seems that it is more likely an additive effect than a synergistic effect. Especially, the gain of survival through the WBRT led to more anti-PD1 treatments. Thus, the statement that it is a synergistic effect is not clearly shown by these data. This should be at least discussed.

Point 2: Measurement Response Rate without using existing definitions

The calculation of the overall response rate (ORR) should be translational to human. Therefore, I

suggest to use already existing response criteria of PD, SD, PR, and CR (e.g. RANO or RECIST). The currently used definitions are different from the established ones. Furthermore I would suggest to replace the picture in Figure 6 B (WBRT and PD1) demonstrating a CR. In Figure 6E the group of WBRT and anti-PD1 show exactly 50% of PD at day 28. However, there are n = 9 animals. How calculate you 50% of n=9 animals? In Figure 6 C there is only one line going beyond day 56 in the WBRT + anti-PDL1 group. However, The OS curve include three long term surviving mice of this treatment group. The CTL (without any manipulation) is missing in Figure 6G. Please, include the untreated control.

Point 3: Number of intracranial metastases.

Furthermore, because the intracranial injection model has a significant biological diversity (numbers of BM and location of BM) at least the numbers and not only the volume should be reported. In the current manuscript, there is no reporting about the numbers at baseline.

Taken together, this is a very important study however, the interpretation of the findings should be more discussed.

Referee #2 (Remarks for Author):

In this paper Niesel et al. present data that in a model of brain metastasis there is anti-tumour T cell priming producing anti-tumour T cell responses. These T cells are ineffective at controlling initial tumour growth and anti-PD1 monotherapy is ineffective at treating these mice. Radiotherapy leads to extended survival and there is a synergistic effect when combined with anti-PD1 therapy. After a period this effect is lost and the authors posit that this is due to myeloid cell mediated immune suppression. Much of the data is very convincing in this manuscript but there is a major concern about the interpretation of the data which impacts the findings of the paper significantly and would require significant changes to address. There are also some minor issues regarding some of the data which can be easily fixed.

Major concern

While the data establishing the model and the immune response is convincing the final conclusions about myeloid cells being the mediators of immunosuppression are much less so. This is more of an issue given the title of the paper which suggests that this final point is the main point the authors wish to make. The T cell stimulation assay used in figure 7J-K is insufficient to carry this point. Firstly the read out of CD69+ or CD69- is very broad and needs to be complemented by other read outs e.g. T cell proliferation, cytokine production, markers of differentiation. Also CD69 levels are highly influenced by number of cell divisions and so are confounded when simply looking at positivity. Furthermore it is unclear which statistical test was used in 7K to determine that the T cells exposed to BMDM + tumour conditioned media have significantly reduced CD69 levels - the tests listed in the legend are t-tests or ANOVA but if this is a t-test which populations were compared or was it a 1 sample t-test compared to the set line of 1? If it was an ANOVA across all populations what was the post-test to specify this condition? Finally there are issues with using t-tests for groups of 3, a non-parametric test would be more appropriate. However, from looking at the graph it would appear that it would be unlikely that BMDM alone vs BMDM + tumour media would be significantly different using this more appropriate test weakening the suggestion that tumour cells are conditioning BMDM to become suppressive. Furthermore in discussing S7B the authors note that anti-PD1 reverses the effect of immune suppression by the BMDM + tumour conditioned media. If this is the mechanism by which the myeloid cells are mediating their

immunosuppression it is unclear why treating the mice with anti-PD1 in the model system does not overcome this immunosuppression. This suggests that either the myeloid cells are suppressing by some other mechanism (and so the in vitro assay is not informative) or the myeloid cells are not the major contributors to immunosuppression in vivo. This relates to the point that consistently the only read out of something being the contributor to immunosuppression is expression of PD-L1 which is similarly complicated by the lack of response to anti PD-1.

In summary to draw the conclusion in the title of the paper that myeloid cell-mediated immune suppression blunts immunoradiotherapy in this model more work is needed both in vitro and this then needs to be translated in vivo through targeting of myeloid cells.

Minor concerns

1. In figure 2A and 2B the authors refer to dendritic cells being stained in sections and in flow cytometry - the markers used show cDC2 not cDC1 (which are CD11b- and (mostly) DCIR2-). cDC2 tend to be more important for CD4 T cell priming and cDC1 for CD8 priming. Since the lack of change in DC is contrasted with the increase in CD8 T cells this is a bit misleading and so should be clarified. It could also be improved by repeating the flow cytometry with a more comprehensive panel to characterise both cDC1 and cDC2.

2. Again relating to flow cytometry, in figure 2E it appears approx. only 50% of CD3+ T cells are expressing either CD4 or CD8 meaning that approx. 50% are something else, it would be good to understand what these cells are as this population is larger than both populations of T cells subsequently characterised.

3. It should be clarified that the cervical lymph nodes examined were the deep cervical nodes which drain the brain rather than the superficial cervicals which drain the auricle.

4. Would it be possible to clarify why anti-PD1 wasn't combined with anti-CTLA4 therapy in this model since this showed good response in Taggart et al. in their B16 model? Would this overcome the issues of acquired resistance in this model too?

Referee #3 (Comments on Novelty/Model System for Author):

The model system used in this study is well-suited to investigate immune suppression and possible strategies to modulate these responses in metastatic cancer. My only concern about the technical quality is the robustness of the data in Figure 2E. According to the figure legend, this experiment was only done once and the results seem to be quite variable. Since one of the major mechanistic conclusions drawn by the authors is that RT increases the number of CD8+ T cells within metastatic lesions, more robust flow cytometry data would make this more persuasive. I do understand the current COVID situation may make these additional experiments difficult, so another option would be to soften the language surrounding this conclusion. I have included this concern in the remarks to the author.

Referee #3 (Remarks for Author):

In "Myeloid cell-mediated immune suppression blunts efficacy of radio-immunotherapy in brain metastasis," the authors investigate the efficacy of radiotherapy (WBRT) combined with immune-checkpoint blockade (ICB) in a syngeneic breast-to-brain metastasis model. Due to the highly immune suppressive environment of the brain and the low immunogenicity of breast-to-brain

metastases, this model is a tractable system to examine whether radiotherapy can be used as an immune modulatory agent to sensitize metastases for immunotherapy. The authors find that T cells infiltrate breast-to-brain metastases and undergo clonal expansion suggesting previous T cell activation and a potential avenue to activate anti-tumor responses. However, upon T cell depletion, onset and progression of metastases remain unchanged, suggesting an immunosuppressive tumor microenvironment. Indeed, the authors observe PD-1/PD-L1 expression on both immune cells and tumor cells within metastatic lesions. To determine whether checkpoint blockade could lift this immunosuppression, the authors utilize WBRT, PD-1 blockade, or the combination of both, and demonstrate prolonged survival and reduced tumor progression. Nonetheless, this response is not sustained due to induction of compensatory immune suppressive mechanisms that limit the efficacy of ICB, likely through myeloid cell-mediated suppression.

The potential synergy between WBRT and checkpoint blockade is well-demonstrated by this study and suggests a possible avenue to activate T cells within the tumor microenvironment. Furthermore, the demonstration that additional immune-modulatory treatments aimed at the suppressive functions of myeloid cells will be required for durable anti-tumor responses is an important finding with clinical relevance. However, the data showing that the mechanism of radiotherapy-induced immune modulation works via increased CD8+ T cell number and function is not as convincing as it could be. If these findings are robust in repeat experiments and T cell functional data can be added, the mechanisms of the proposed model would be more persuasive. Overall, this study is important for the field and adds insight into which aspects of the immune response to metastasis should be targeted for increased therapeutic efficacy and should be revised with attention to the comments listed below.

Specific points to address:

Figure 2D: I assume that this is quantification of images as in 2C? The legend does not make this clear.

Figure 2E: Have these experiments been repeated? The variability here seems high so differences in T cell percentages (especially the increase in CD8+ T cells) would be more convincing if these differences are consistent between multiple experiments (and/or with more data points).

Figure 3: Were these clonal analyses performed with CD4+ and CD8+ T cells separately, or only bulk as in this figure? It would be interesting to include separate analyses (especially for CD8+ T cells) if the authors have these data already collected.

Figure 7: Since the combination therapy-treated mice survive much longer, have the infiltrates of these lesions ever been analyzed in a time course (as opposed to the trial endpoint)? It would be interesting to see the extent of the changes in PD-1/PD-L1 expression in the time points between experimental endpoint with the other groups (~28 days?) vs. when the mice succumb to disease. However, the reviewer understands that these experiments require a lot of time/mice so this is not a necessary addition.

Figure 7K: The changes in CD69 expression are subtle in response to culture with BMDMs preconditioned with tumor-conditioned media. Were any other markers of T cell activation or function used in these experiments (ie. GzmB, etc)? Additional markers or a functional readout would better support the argument that monocyte-derived macrophages in the tumor microenvironment are potentially suppressive to T cells.

Summary and general remarks

It is our pleasure to resubmit our manuscript on the effects of radio-immunotherapy in breast-to-brain metastasis. We would like to sincerely thank the editor and the referees for acknowledging the impact of the study and for their thoughtful and constructive comments that helped to improve our manuscript. In the revised manuscript, we have addressed all reviewer's comments and have:

1. Added data new data sets including RNAseq data sets of tumor-associated myeloid and lymphoid cells in comparison to their normal cellular counterparts from brain and blood to provide a more detailed characterization of the respective cell types in brain metastases (**new Fig 1D-I and Fig EV1, new Fig 3G**). The newly added data supports our results that myeloid cells play key roles in modulating tumor-associated inflammation and that especially monocyte-derived macrophages are critical regulators of T activity in brain metastases. This further allowed us to query the data set for expression of genes of interest such T cell activation and exhaustion marker in tumor-infiltrating CD4+ and CD8+ T cells (**new Fig 2H**) as well as expression a panel of co-regulatory factors in myeloid and lymphoid populations (**new Fig 3E and F**) to address specific comments from the reviewers.
2. Included multiplexed histology to provide insight into the spatial distribution of myeloid and lymphoid subpopulations in brain metastases and across treatment groups (**new Fig 3H and Fig EV5**).
3. Included more detailed characterization of the model with respect to the presence of extracranial tumors by MRI and histology in representative organs (**new Appendix Fig 1**).
4. Extended and improved flow cytometry panels for more detailed characterization of the cellular composition of dendritic cells and double negative T cells (**new Fig 1A-C and 4G-J**).
5. Extended and improved the in vitro T cell activation assay by including the analyses of additional markers (Granzyme B and Interferon gamma) and increased the number of replicates (**new Fig 7K**).
6. Modified the wording to tone down interpretation of specific results.
7. Changed the title to better reflect the data presented in the manuscript
8. Included further points in the discussion such as comparison of different radiation regimen (WBRT vs. SRS) and other non-myeloid cell types that have been associated with immune suppression in CNS tumors.

The RNAseq and TCRseq data have been deposited and can currently be accessed via the following links:

RNAseq: <https://www.ncbi.nlm.nih.gov/geo/query/acc.cgi?acc=GSE164049>

TCRseq: clients.adaptivebiotech.com

Addition of the new data sets led to a re-organization of individual figures as well as manuscript sections to integrate the new data into the manuscript. A detailed overview of the modifications within the revised manuscript can be found in the point-to-point response below. We hope that the addition of new data and interpretation of the results in the revised version addresses the concerns raised by the reviewer.

***** Reviewer's comments *****

Referee #1 (Comments on Novelty/Model System for Author):

This is a technically very complex investigation with many valuable results. In particular, the experiments underline the importance of the myeloid compartment in brain metastasis. In addition, the data underline clinical observations and provide some biological explanations. In particular, the very elaborate models, the imaging, the irradiation and the technical processing are very impressive. The weakness of the manuscript is more the interpretation of the results and the discussion of the limitations. The intracranial injection model has a very high biological variance and, in particular, extracranial micro-metastasis formations cannot be excluded. For example, no reference is made to this point. But whether extracranial micrometastases are present could be of enormous importance for the immune reaction (ICB). There are further important limitations of this study (see text below). From my point of view, the limitations at least should be discussed. In view of this, the discussion of the possible signaling pathways of the myeloid signaling pathway should be more pronounced and the role the glia, e.g. Astrocytosis almost missing. Astrocytes are also important candidates for the immunosuppressive reaction of the brain tissue. For these reasons, I recommend at least a restructuring of the discussion and abstract, especially synergism must be discussed (see below).

Another important point is also not mentioned. Usually solitary metastases are treated with SRS and not WBRT. SRS and WBRT definitely have different biological and clinical effects. This should at least be mentioned. Other parts of the discussion can be significantly shortened.

Response: We would like to thank the referee for acknowledging the complexity and impact of our study. We acknowledge the concern raised with respect to the interpretation of the results. As explained in more detail in the point-to-point answers below we included new data sets to address the points raised by the reviewer and/or modified the discussion of the mentioned points. E.g. we included a more detailed characterization of the extracranial tumor burden in the 99LN-BrM model based on abdominal MRI and histological evaluation of extracranial tumor burden. Moreover, we added the discussion point on possible biological differences between SRS and WBRT. We emphasized the importance of a direct comparison of those different treatment modalities to define the optimal regimen to sensitize brain metastases to immune check point blockade by adding the following

information in the discussion: “Given recent clinical data, comparison of WBRT and stereotactic radiosurgery (SRS) will be of critical importance to define optimal treatment modalities to induce immune modulatory effects that synergize with ICB in brain metastases (Kotecha et al, 2019; Lehrer et al, 2018).”

Point 1: Measurement of extracranial tumor volume is missing and synergistic effect

The introduction cited the very important findings of Taggart et al. 2018. These findings underlined the impact of the extracranial Tumor for effective intracranial ICB. Taggart et al. demonstrated that without extracranial tumor tissue, there will be no intracranial response of ICB. Here, all the data derive from an intracardiac injection of the 99LN BrM metastasis model. Despite the 99LN BrM brain seeking cell line used for the experiments, it is unlikely that there is no extracranial metastasis growth at all (e.g. micro-metastasis in the bone marrow). The current manuscript do not report, how the extracranial metastasis load is measured. Also both citations (Bowman et al 2016 and Chae et al 2019) did not describe the frequency of extracranial (micro)-metastasis formation of the 99LN BrM model. However, to interpret all the immunological-findings (ICB) in respect to Taggart et al, it would be important to know, the frequency, and volume of the extracranial tumor mass. In particular, it could be one important difference between non-responder and responder to ICB (Figure 6). This is also important for the statement that radiotherapy is really synergistic. If you only add the differences between the OS data:

WBRT 42 days - Iso-Type control 28 days =difference of 14 days

aPD1 39 days - Iso-Type control 28 days =difference of 11 days

An additive effect would be around 28 days (control) + 14 days (difference WBRT) + 11 days (difference ICB) = 53 days

Interestingly the Median OS of WBRT + anti PD1 is 52 days. Thus, it seems that it is more likely an additive effect than a synergistic effect. Especially, the gain of survival through the WBRT led to more anti-PD1 treatments. Thus, the statement that it is a synergistic effect is not clearly shown by these data. This should be at least discussed.

Response: We would like to thank the reviewer for mentioning the important aspect of a possible influence of extracranial tumor lesions on response rates to ICB monotherapy or radio-immunotherapy. Unfortunately, we cannot retrospectively evaluate the extent of the extracranial tumor load in the mice used for the survival analyses to correlate response rates to the presence of extracranial tumors. To better estimate the extent of extracranial tumor burden in the 99LN-BrM model, we performed abdominal MRI using an independent cohort and performed histological evaluation of different organs including bones, spleen, liver and lungs. We did not detect extracranial tumors in bones, spleen or liver in the analyzed mice. The only extracranial tumors were detected in one mouse that developed lung metastases. The data is now included in Appendix data figure S1. In future studies, we will include analyses of the extracranial tumor load at trial endpoint to be able to directly correlate response rates to potential extracranial tumor load to further address the question whether differences in T cell infiltration and response rates in the 99LN-BrM breast-to-brain metastases model are affected by extracranial tumors.

Figure R1.1 Overview of intracranial and extracranial tumor lesions in the 99LN-BrM model by MRI and histology. Data can be found as **Appendix Fig S1** in the revised manuscript.

We agree with the reviewer that the calculation of median survival time across the experimental groups rather argues for additive effects of the combination of anti-PD-1 and WBRT. However, analysis of the tumor growth kinetics of the anti-PD-1 monotherapy did not show reduction of tumor progression (e.g. Fig 6C and Fig EV4B), suggesting that effects on tumor growth in the combined treatment group results from synergy between WBRT

and anti-PD1. In order to prevent improper usage of the terms synergistic or additive effects, we changed the wording and only refer to “further improved” effects to describe the effects of radio-immunotherapy in the revised manuscript when referring to our results.

Point 2: Measurement Response Rate without using existing definitions

The calculation of the overall response rate (ORR) should be translational to human. Therefore, I suggest to use already existing response criteria of PD, SD, PR, and CR (e.g. RANO or RECIST). The currently used definitions are different from the established ones.

Response: We thank the reviewer for suggesting to use already existing response criteria of PD, SD, PR and CR such as RANO and RECIST criteria that are usually used to classify treatment responses in patients. Those criteria are based on tumor diameter while our measurements are based on tumor volume. We therefore chose to use previously defined criteria for brain tumor mouse models that were also based on tumor volume. We now clearly stated in the manuscript that we stratified responses to treatment into progressive disease (PD; change in BrM volume > +40%), stable disease (SD; change in BrM volume between -65% and +40%), partial response (PR; change in BrM volume >-65%) and complete response (CR; change in BrM volume = -100%) based on criteria previously proposed for a mouse brain tumor model (Aslan et al, 2020)

Ref: Aslan K, Turco V, Blobner J, Sonner JK, Liuzzi AR, Nunez NG, De Feo D, Kickingereeder P, Fischer M, Green E et al (2020) Heterogeneity of response to immune checkpoint blockade in hypermutated experimental gliomas. Nat Commun 11: 931

Furthermore I would suggest to replace the picture in Figure 6 B (WBRT and PD1) demonstrating a CR. In Figure 6E the group of WBRT and anti-PD1 show exactly 50% of PD at day 28.

Response: Images were chosen to depict a representative course of disease for each treatment group. We believe that the presentation of an animal showing SD is reflecting disease progression best in this group best. Please note that this mouse does not represent a mouse showing CR. The volume that was measured for this representative lesion included the area as indicated in the images below and did not exclude the area with lower contrast enhancement.

However, there are n = 9 animals. How calculate you 50% of n=9 animals? In Figure 6 C there is only one line going beyond day 56 in the WBRT + anti PDL1 group. However, The OS curve include three long term surviving mice of this treatment group. The CTL (without any manipulation) is missing in Figure 6G. Please, include the untreated control.

Response: We would like to thank the reviewer for noticing the missing data point in Fig 6C for one of the long-term survivors in the anti-PD-1+WBRT treatment group. The reason for the missing data point was due to technical issue with the MRI so that the d56 time point could not be acquired for this mouse. To achieve higher consistency of presentation between both graphs (Fig 6C and 6E), we now included the d63 time point to depict progression of this animal as well. In Fig 6E, there are 2 animals surviving beyond d56 in the anti-PD-1 treatment group. The raw data for the graphs is provided in the Data Source file.

The percentage of animals showing PD, SD, PR or CR are calculated based on numbers of animals still alive at each time point. Mice were included at earlier time points even though the mice were not alive at later time points. As indicated in the survival curve, not all mice died from brain metastases but had to be sacrificed due to other reasons e.g. elaborate breathing. Those animals are indicated with a tick in the survival curves. E.g. for the PD-1 +WBRT treatment group, 3 animals had to be sacrificed due to symptoms not correlated to BrM on d12, d24 and d112, while the intracranial tumor burden was very low. The figure legend now states that the indicated number of animals refers to the time of treatment start. For more precise presentation of the actual mouse numbers at each time point presented in Fig 6E, we now included the mouse numbers directly within the columns. The detailed description of the percentage of mice in each category has now been shortened for conciseness to accommodate the additional data sets in the revised version within the allowed character limits.

The graph in Fig 6G represents data for the four treatment groups enrolled in the in vivo trials. In this setting, mice treated with WBRT received injections of the isotype control IgG2a for direct comparison with the Isotype group and the groups that received anti-PD-1 treatment. Based on our experience with the 99LN-BrM models across multiple projects that include different control conditions such as untreated control mice, isotype- or carrier-treated animals, we are confident that the injection of the isotype control as described herein does not affect tumor progression compared to untreated animals. Given the experimental design of the survival trial we believe that

comparison to the isotype control group is sufficient. To further clarify that WBRT-treated animals also received injections of the isotype control, we now included the following description in the text in addition to the illustration in the overview of the experimental design. "Mice from the α PD-1 and α PD1+WBRT group were treated with α PD-1 (250 ug/d) every third day throughout the trial period. Mice from the isotype and WBRT group received injections of the isotype control (IgG2a) (Fig 6A)."

Point 3: Number of intracranial metastases.

Furthermore, because the intracranial injection model has a significant biological diversity (numbers of BM and location of BM) at least the numbers and not only the volume should be reported. In the current manuscript, there is no reporting about the numbers at baseline.

Taken together, this is a very important study however, the interpretation of the findings should be more discussed.

Response: Brain metastases are induced via intracardiac injection of variants that were selected for brain tropism. Indeed, intracardiac injection models have a significant biological diversity in terms of number and location of brain metastases. To provide further details about the number of brain metastases at treatment start, we included a new figure panel in Fig EV4A to present the number and tumor volume in each treatment group demonstrating equal distribution of number and size across the treatment groups at baseline.

Figure R1.3. Quantification of the number and volume of brain metastases across the different groups before treatment start. Data can be found in **Fig EV4A** in the revised manuscript.

Referee #2 (Remarks for Author):

In this paper Niesel et al. present data that in a model of brain metastasis there is anti-tumour T cell priming producing anti-tumour T cell responses. These T cells are ineffective at controlling initial tumour growth and anti-PD1 monotherapy is ineffective at treating these mice. Radiotherapy leads to extended survival and there is a synergistic effect when combined with anti-PD1 therapy. After a period this effect is lost and the authors posit that this is due to myeloid cell mediated immune suppression. Much of the data is very convincing in this manuscript but there is a major concern about the interpretation of the data which impacts the findings of the paper significantly and would require significant changes to address. There are also some minor issues regarding some of the data which can be easily fixed.

Response: We would like to thank the reviewer for the positive evaluation and the constructive comments. We acknowledge the concerns raised by the reviewer with respect to the interpretation of the data. In order to address these points we included additional data to support our interpretation and/or changed the wording to prevent overinterpretation of individual findings.

Major concern

While the data establishing the model and the immune response is convincing the final conclusions about myeloid cells being the mediators of immunosuppression are much less so. This is more of an issue given the title of the paper which suggests that this final point is the main point the authors wish to make.

Response: We acknowledge the concern raised by the reviewer that the statement of the original manuscript title was not completely supported by the data presented. We therefore propose to change the title to: "The immune microenvironment affects efficacy of radio-immunotherapy in brain metastasis"

The T cell stimulation assay used in figure 7J-K is insufficient to carry this point. Firstly the read out of CD69+ or CD69- is very broad and needs to be complemented by other read outs e.g. T cell proliferation, cytokine production, markers of differentiation. Also CD69 levels are highly influenced by number of cell divisions and so are confounded when simply looking at positivity.

Response: We apologize if we created the impression that the statement on the role of myeloid cells in mediating immune suppression was solely based on data presented in Fig 7J-K in the original submission. To further support our conclusion we:

1) Added new data sets including RNAseq data to provide unbiased insight into myeloid and lymphoid populations in the 99LN-BrM model. The new data is included in Figure 1D-I plus supportive data in Fig EV1 and Fig. 3.

Figure R2.1. RNAseq data of tumor-associated myeloid and lymphoid populations compared to their normal cellular counterparts from normal brain and peripheral blood. Data can be found in **Fig 1D-I** in the revised version.

Figure R2.2 Functional gene annotation of altered cellular pathways in TAM-MG vs. TAM-MDM. Data can be found in **Fig 3G** in the revised version.

Furthermore it is unclear which statistical test was used in 7K to determine that the T cells exposed to BMDM + tumour conditioned media have significantly reduced CD69 levels - the tests listed in the legend are t-tests or ANOVA but if this is a t-test which populations were compared or was it a 1 sample t-test compared to the set line of 1? If it was an ANOVA across all populations what was the post-test to specify this condition? Finally there are issues with using t-tests for groups of 3, a non-parametric test would be more appropriate. However, from looking at the graph it would appear that it would be unlikely that BMDM alone vs BMDM + tumour media would be significantly different using this more appropriate test weakening the suggestion that tumour cells are conditioning BMDM to become suppressive.

Response: We apologize if the description of the statistical test that was used to analyze the data presented in 7K of the original submission was not clear. We used paired t-test for the statistical analyses of the data presented in Fig 7J and K. We updated the information provided in the figure legend:

“Data information: Data in (A-C, E+F) is represented as scattered dot plot with line at mean \pm SD. Data in (D+G+H) is represented as stacked columns \pm SD. Data in (J+K) is represented as scattered dot plot with line at median. Dotted line depicts baseline T cell activation. P-values were obtained by unpaired t test in (C and F), ordinary two-way ANOVA in (D, G and H) and paired t test in (J and K) with * P <0.05 and ** P <0.01.”

And in the main text: T cell activation was measured by flow cytometry of the activation marker CD69, IFNg and Grzmb after 24h (Fig 7I). 99LN-BrM tumor cells alone did not significantly change T cell activity based on CD69, IFNg and Grzmb protein level (Fig 7J and K). Across all analyzed conditions, BMDM showed the highest capacity to reduce CD69 and IFNg on T cells compared to baseline T cell activation.

Please note that the ordinary two-way ANOVA test was performed for Fig 7D, G and H to evaluate differences in the distribution of cell types within and across groups. Data on post-hoc tests have been included in the Data Source file.

Based on the reviewers suggestion, we extended the markers that were analyzed in the in vitro T cell activity assay and included analyses of Granzyme B and IFNg levels with n=4-5 replicated per condition. The data has been included in Fig 7K.

Figure R2.3 In vitro T cell activation assay based on IFNg and Grzmb levels in CD4 and CD8+ T cells across different experimental conditions. Data can be found in **Fig 7K** of the revised manuscript.

Furthermore in discussing S7B the authors note that anti-PD1 reverses the effect of immune suppression by the BMDM + tumour conditioned media. If this is the mechanism by which the myeloid cells are mediating their immunosuppression it is unclear why treating the mice with anti-PD1 in the model system does not overcome this immunosuppression. This suggests that either the myeloid cells are suppressing by some other mechanism (and so the in vitro assay is not informative) or the myeloid cells are not the major contributors to immunosuppression in vivo. This relates to the point that consistently the only read out of something being the contributor to immunosuppression is expression of PD-L1 which is similarly complicated by the lack of response to anti PD-1. In summary to draw the conclusion in the title of the paper that myeloid cell-mediated immune suppression blunts immunoradiotherapy in this model more work is needed both in vitro and this then needs to be translated in vivo through targeting of myeloid cells.

Response: We would like to thank the reviewer for discussing this important point. We further analyzed the effect of PD-1 inhibition in the in vitro T cell activation assay. This analysis revealed that the addition of anti-PD1 resulted in a significant increase of GrzmB levels in CD4 T cells in the BMDM pre-stimulated+99LN-BrM condition and a slight increase in IFNg levels in CD8+ T cells in the same condition. To better reflect the observed effect, we toned down the interpretation of the data and state now:

‘The addition of aPD-1 resulted in a slight increase of IFNg and Grzmb in CD4⁺ and CD8⁺ cells and a significant increase in Grzmb levels on CD4+ T cells. Interestingly, effects of PD-1 inhibition were only observed for BMDM after preconditioning with TuCM and co-culture with 99LN-BrM cells (Fig 7K), whereas the addition of aPD-1 to conditions with EOC2 cells did not lead to changes in IFNg and Grzmb levels. This suggest that T cell suppression by BMDM is at least in part mediated via the PD-1 – PD-L1 axis, while T cell suppression by EOC2 is likely mediated by alternative pathways.’

Moreover, we queried the newly added RNAseq data set for the expression of co-regulatory factors on myeloid cells and the corresponding receptors on T cells to gain further insight into potential alternative pathways that lead

to T cell suppression. We believe that this data represents a valuable source for the identification of additional immune checkpoints that regulate T cell activation in brain metastases and potentially compensate in response to PD1 inhibition. This data has now been added to Fig 3E and F. While further functional validation of those findings are beyond the scope of the present study, we seek to further investigate those questions in follow-up studies.

Figure R2.4 Expression of different genes of interest representing genes involved in antigen presentation as well as co-regulatory factors and receptors in myeloid and lymphoid cells. Data can be found in **Fig 3E and F** of the revised manuscript.

Minor concerns

1. In figure 2A and 2B the authors refer to dendritic cells being stained in sections and in flow cytometry - the markers used show cDC2 not cDC1 (which are CD11b- and (mostly) DCIR2-). cDC2 tend to be more important for CD4 T cell priming and cDC1 for CD8 priming. Since the lack of change in DC is contrasted with the increase in CD8 T cells this is a bit misleading and so should be clarified. It could also be improved by repeating the flow cytometry with a more comprehensive panel to characterise both cDC1 and cDC2.

Response: We would like to thank the reviewer for noticing the mistake in labeling of the axis. The original data presented in Fig 2B of the submitted manuscript was based on the marker combination CD83+CD11c+ as illustrated in the gating strategy. Unfortunately, the axis in Fig2B was labeled with CD11b instead of CD11c. We sincerely apologize for this mistake. Yet, we agree that the point mentioned by the reviewer on the importance of distinguishing DC subpopulations given their role in T cell priming, is critical. We therefore repeated the DC analyses employing an alternative gating strategy for DC in brain tumors as previously proposed by Mrdjen et al. using MHCII, CD11c, CD11b and CD24. Further information on the FACS panels can be found in Appendix Fig S2. This analysis revealed that most DC in 99LN-BrM are cDC2 (appr. 50% of the DC population) followed by cDC1 (appr. 10% of the DC population) as well as a remaining population that contains pDC and has been denoted as pDC/other. The data presenting the proportion of DC populations, were included in a new graph in Fig 1C. Moreover, the effect of WBRT on the DC populations was evaluated. This data is now included in Fig 4G-J. We did not detect significant effects of WBRT on DC populations.

Ref: Mrdjen D, Pavlovic A, Hartmann FJ, Schreiner B, Utz SG, Leung BP, Lelios I, Heppner FL, Kipnis J, Merkler D et al (2018) High-Dimensional Single-Cell Mapping of Central Nervous System Immune Cells Reveals Distinct Myeloid Subsets in Health, Aging, and Disease. *Immunity* 48: 599

Figure R2.6. New FACS panel to further classify dendritic cell and T cell populations. The data can be found in **Fig 1C** of the revised manuscript.

Figure R2.7. New FACS panel to analyze DC populations in response to WBRT. The data can be found **Fig 4G-J** in the revised manuscript.

2. Again relating to flow cytometry, in figure 2E it appears approx. only 50% of CD3+ T cells are expressing either CD4 or CD8 meaning that approx. 50% are something else, it would be good to understand what these cells are as this population is larger than both populations of T cells subsequently characterised.

Response: We agree with the reviewer, that brain metastases harbor a high proportion of double negative T cells. In order to provide further information on the identity of the double negative T cells, we performed further flow cytometric analyses. The T cell flow panels were extended by the addition of the markers DX5 and gd-TCR to quantify the proportion of NKT cells and gd T cells within the double negative population. Further information on the FACS panels can be found in **Appendix Fig S2**. These analyses revealed a prominent proportion of NKT cells in the double negative population and a smaller proportion of gd T cells. Although this analysis allowed us to determine the identity of a considerable amount of the double negative population, there remains a double negative T cell population that we were not able to further stratify (now denoted as other DN cells). The data presenting further subpopulations within the T cell population have now been included in the newly created **Fig 1C (Figure R2.6 see above)**. Future analyses employing fully unbiased approaches will hopefully help to further characterize this population.

3. It should be clarified that the cervical lymph nodes examined were the deep cervical nodes which drain the brain rather than the superficial cervicals which drain the auricle.

Response: We would like to thank the reviewer for raising this point. Although recent reports indicate that deep CLN are the major lymphnodes draining the CNS, routes of exit are not fully elucidated yet. For example, data recently published by Louveau et al. demonstrate that lymphatic exit routes from the CNS might split and drain into dCLN and superficial CLN (sCLN). Moreover, models used for the detailed investigation of lymphatic exit routes might not represent the situation found in brain tumors, which heavily modulate and disrupt their microenvironment. Therefore, we harvested and pooled CLN, including dCLN and sCLN to prevent potential loss of TCR-clones. CLN harvest was performed consistent between mice and conditions. In future experiments, it will be crucial to further investigate lymphatic exit routes in BrM and in line with this analyze TCR clonality in dCLN and sCLN separately.

Reference: Louveau A, Herz J, Alme MN, Salvador AF, Dong MQ, Viar KE, Herod G, Knopp J, Setliff J, Lupi AL, et al (2018) CNS lymphatic drainage and neuroinflammation are regulated by meningeal lymphatic vasculature. *Nat Neurosci* 21: 1380–1391

4. Would it be possible to clarify why anti-PD1 wasn't combined with anti-CTLA4 therapy in this model since this showed good response in Taggart et al. in their B16 model? Would this overcome the issues of acquired resistance in this model too?

Response: We thank the reviewer for this excellent suggestion. Indeed, preclinical and clinical data indicate that combination of checkpoint inhibitors increases response rates and might help to overcome the issue of acquired resistance in this model. While the focus of the present study was on effects of radiotherapy on the efficacy of inhibition of one of the prominent immune checkpoint molecules critical for T cell activity during the effector phase at the tumor site, we plan to evaluate the efficacy of blockade of additional checkpoints such as CTLA4 or other co-regulatory factors that we found to be highly expressed on myeloid cells, in the future.

Referee #3 (Comments on Novelty/Model System for Author):

The model system used in this study is well-suited to investigate immune suppression and possible strategies to modulate these responses in metastatic cancer. My only concern about the technical quality is the robustness of the data in Figure 2E. According to the figure legend, this experiment was only done once and the results seem to be quite variable. Since one of the major mechanistic conclusions drawn by the authors is that RT increases the number of CD8+ T cells within metastatic lesions, more robust flow cytometry data would make this more persuasive. I do understand the current COVID situation may make these additional experiments difficult, so another option would be to soften the language surrounding this conclusion. I have included this concern in the remarks to the author.

Response: We would like to thank the reviewer for the positive evaluation of our study and the understanding of possible issues to complete experiments due to the current COVID situation. Despite the current limitations, we aimed to provide additional experimental data to address the majority of the comments.

Referee #3 (Remarks for Author):

In "Myeloid cell-mediated immune suppression blunts efficacy of radio-immunotherapy in brain metastasis," the authors investigate the efficacy of radiotherapy (WBRT) combined with immune-checkpoint blockade (ICB) in a syngeneic breast-to-brain metastasis model. Due to the highly immune suppressive environment of the brain and the low immunogenicity of breast-to-brain metastases, this model is a tractable system to examine whether radiotherapy can be used as an immune modulatory agent to sensitize metastases for immunotherapy. The authors find that T cells infiltrate breast-to-brain metastases and undergo clonal expansion suggesting previous T cell activation and a potential avenue to activate anti-tumor responses. However, upon T cell depletion, onset and progression of metastases remain unchanged, suggesting an immunosuppressive tumor microenvironment. Indeed, the authors observe PD-1/PD-L1 expression on both immune cells and tumor cells within metastatic lesions. To determine whether checkpoint blockade could lift this immunosuppression, the authors utilize WBRT, PD-1 blockade, or the combination of both, and demonstrate prolonged survival and reduced tumor progression. Nonetheless, this response is not sustained due to induction of compensatory immune suppressive mechanisms that limit the efficacy of ICB, likely through myeloid cell-mediated suppression.

The potential synergy between WBRT and checkpoint blockade is well-demonstrated by this study and suggests a possible avenue to activate T cells within the tumor microenvironment. Furthermore, the demonstration that additional immune-modulatory treatments aimed at the suppressive functions of myeloid cells will be required for durable anti-tumor responses is an important finding with clinical relevance. However, the data showing that the mechanism of radiotherapy-induced immune modulation works via increased CD8+ T cell number and function is not as convincing as it could be. If these findings are robust in repeat experiments and T cell functional data can be added, the mechanisms of the proposed model would be more persuasive. Overall, this study is important for the field and adds insight into which aspects of the immune response to metastasis should be targeted for increased therapeutic efficacy and should be revised with attention to the comments listed below.

Response: We agree with the reviewer that repetition of key experiments and increasing the number of analyzed animals would support our findings. We therefore repeated key experiments to include more animals. Addition of animals further supported our findings presented in the original submission. This data has now been included in Fig 4 O-R (see below). To provide further T cell functional data, we now included a new RNAseq data set with expression data of CD4+ and CD8+ T cells with functional annotation of differentially expressed genes. This data has been added to Fig 1E, H and I (see below) to display enriched pathways in tumor-infiltrating T cells compared to normal blood lymphocytes. Moreover, we queried this data set for the expression of T cell activation and exhaustion marker and observed induction of exhausted T cell phenotype. This analysis has now been added in Fig 2H. We further analyzed the expression of stimulating or inhibitory co-regulatory receptors on T cells to gain insight into T cell exhaustion phenotypes.

Figure R3.1. Expression of a panel of T cell activation and exhaustion marker in CD4+ and CD8+ T cells isolated from brain metastases or peripheral blood of tumor-free animals. The data can be found in **Fig 2H** of the revised manuscript.

Specific points to address:

Figure 2D: I assume that this is quantification of images as in 2C? The legend does not make this clear.

Response: Yes, Fig 2D of the original submission was the quantification of images as in 2C. This data has now been moved to Fig 4K (Histology) and 4L-N (Quantification). The data are now adjacent to each other in the new figure that it is better visible that the data in L-N is the quantification of 4K.

Figure 2E: Have these experiments been repeated? The variability here seems high so differences in T cell percentages (especially the increase in CD8+ T cells) would be more convincing if these differences are consistent between multiple experiments (and/or with more data points).

Response: Yes, as stated above, we repeated those experiments in independent cohorts and confirmed our initial finding that WBRT increases the amount of CD8+ T cells. The revised data can be found in Fig 4O-R.

Figure R3.2. Repetition of the quantification of T cells in brain metastases in response to WBRT. The data can be found in **Fig 4O-R** of the revised manuscript.

Figure 3: Were these clonal analyses performed with CD4+ and CD8+ T cells separately, or only bulk as in this figure? It would be interesting to include separate analyses (especially for CD8+ T cells) if the authors have these data already collected.

Response: The analysis was performed on bulk populations as this allowed us to gain further information on e.g. % estimated T cells in BrM with different volume. We agree with the reviewer that separate analyses of CD4+ and CD8+ T cells would have also provided important information on potential differences in clonal expansion. While we cannot provide this data by TCR sequencing, we observed an enrichment of pathways associated with cell cycle, cell division, mitotic cell cycle process etc in CD8+ T cells based on RNAseq data suggesting proliferation particularly in the CD8+ T cell population. This data has been included in Fig 11.

Figure R3.3 Functional annotation of differentially expression genes in CD4+ and CD8+ T cells isolated from brain metastases lesions compared to normal blood lymphocytes. Displayed are pathways that are up- or downregulated in BrM-associated T cells. The data can be found in **Fig 1H and I** of the revised manuscript.

Figure 7: Since the combination therapy-treated mice survive much longer, have the infiltrates of these lesions ever been analyzed in a time course (as opposed to the trial endpoint)? It would be interesting to see they extent of the changes in PD-1/PD-L1 expression in the time points between experimental endpoint with the other groups (~28 days?) vs. when the mice succumb to disease. However, the reviewer understands that these experiments require a lot of time/mice so this is not a necessary addition.

Response: We thank the reviewer for this excellent suggestion. Unfortunately, we were not able to repeat the trial to include analyses of immune infiltrates in a time course given the current limitations due to COVID: However, we plan to include analyses of the immune infiltrate at different time points in response to radio-immunotherapy to address this question in future studies.

Figure 7K: The changes in CD69 expression are subtle in response to culture with BMDMs preconditioned with tumor-conditioned media. Where any other markers of T cell activation or function used in these experiments (ie.

GzmB, etc)? Additional markers or a functional readout would better support the argument that monocyte-derived macrophages in the tumor microenvironment are potentially suppressive to T cells.

Response: We would like to thank the reviewer for the excellent suggestion to improve the in vitro T cell activation assay. Based on the recommendation, we extended our analyses and included protein levels of GzmB and IFN γ in our analyses. We believe that the new data set further supports our statement that monocyte-derived macrophages are critical for modulating T cell activity. The new data has been now added in Fig 7K. Due to the addition of the new analyses, we removed the data we previously presented in Fig S7.

Figure R3.4 In vitro T cell activation assay based on IFN γ and Grzmb levels in CD4 and CD8+ T cells across different experimental conditions. Data can be found in Fig 7K of the revised manuscript.

In addition to the extended T cell activation assay, we included a new data set with RNAseq data to further support our finding that monocyte-derived macrophages in the tumor microenvironment are critical to modulate T cell activity. As for T cells, we included this new data set in Fig 1D, F and G to present functional annotation of differentially expressed genes in tumor-associated vs normal cell types. Moreover, we queried the data set for expression of genes involved in antigen presentation as well as stimulatory and inhibitory co-regulatory molecules on tumor-associated microglia in comparison to tumor-associated monocyte-derived macrophages. These analyses revealed that expression of the majority of the analyzed factors was higher in monocyte-derived macrophages compared to microglia. This was further supported by functional gene annotation showing enriched pathways in tumor-associated microglia vs. monocyte-derived macrophages indicating association of monocyte-derived macrophages with regulation of T cell activity. This data has now been added in Fig 3E, F and G.

Figure R3.5 Functional annotation of differentially expressed genes in TAM-MG and TAM-MDM isolated from brain metastases lesions compared to normal cellular counterparts. Displayed are pathways that are up- or downregulated in BrM-associated myeloid cells. The data can be found in Fig 1F and G of the revised manuscript.

Figure R3.6 Expression of different genes of interest representing genes involved in antigen presentation as well as co-regulatory factors and receptors in myeloid and lymphoid cells. Functional annotation of enriched pathways in TAM-MG vs TAM-MDM. Right panel displays pathways enriched in TAM-MG; left panel displays pathways enriched in TAM-MDM. Data can be found in **Fig 3E-G** of the revised manuscript

Moreover, we performed multiplex immunofluorescence imaging to gain further insight into the spatial distribution of the cell types of interest in this study. These analyses revealed that in BrM, CD4 and CD8+ T cells are localized in closer vicinity to Iba1+TMEM119- cells (representing monocyte-derived macrophages) compared to Iba1+TMEM119+ cells (representing microglia). This spatial organization further supports the statement that monocyte-derived macrophages play a critical role in regulating T cell activity in brain metastases. Figure EV5 shows the corresponding analyses across treatment groups.

Figure R3.7 Multiplex immunofluorescence to analyze the proximity of CD4+ and CD8+ T cells to different cell types within brain metastases including tumor cells and TAM populations. Data can be found in **Fig 3H**.

11th Feb 2021

Dear Dr. Sevenich,

Thank you for the submission of your revised manuscript to EMBO Molecular Medicine. We have now received the enclosed reports from the three referees who re-reviewed your manuscript. As you will see, they are now supportive of publication, and I am therefore pleased to inform you that we will be able to accept your manuscript, once the following minor points will be addressed:

1/ Main manuscript text:

- Please answer/correct the changes suggested by our data editors in the main manuscript file (in track changes mode). Please use this file for any further modification.
- Please move the References after the Conflict of Interest section.
- Material and methods:
 - o Mice: indicate the housing and husbandry conditions, as well as the gender of the mice used in your experiments (this last point should also be indicated in the checklist).
 - o Cells: mention whether the cells were tested for mycoplasma contamination.
- Statistics: Please indicate in the figures or in the legends the exact $n=$ and exact $p=$ values along with the statistical test used. You may provide these values as a supplemental table in the Appendix file.
- Data availability: this section should directly follow the Materials and Methods section. Please note that the datasets have to be publicly available/accessible before acceptance of your manuscript.

2/ Figures and Appendix:

- Dataset EV: for each file, please add a legend in a separate tab.
- Appendix: please rename the figures "Appendix Figure S1" etc. and "Appendix Table S1" etc. Table numbering should start with 1, the numbers indicated in the table of content and in the table titles do not match.
- Please make sure that all figures are referred to in the manuscript text. Figure callouts are missing for Fig 7B, Fig EV1E, Fig EV3A/B, Fig EV4A, panels of Fig EV5, Dataset EV1 and 2.

3/ Thank you for providing Source Data. Please zip them together so as to have one file per figure. The excel file provided for Figure 2 contains tabs that are labeled Figure 4, please check.

4/ Thank you for providing The Paper Explained section. I have slightly modified it to shorten it, please amend as you see fit and incorporate it in the manuscript:

Problem: Response rates to checkpoint inhibitors for poorly immunogenic cancers are low.

Additionally, cancer cells metastasizing to the brain are protected from the adaptive immune system by the immune suppressive tumor microenvironment (TME). Therefore, many brain metastasis (BrM) patients cannot benefit from the advent of immunotherapies and are restricted to standard of care therapies with limited success rates.

Results: We found that cell types essential for successful immunotherapy via PD-1 blockade are present in murine breast cancer derived BrM, including T cells and dendritic cells. T cells in the TME of a breast cancer BrM mouse model clonally expanded, indicating prior T cell activation. A high proportion of T cells expressed PD-1, whereas tumor and myeloid cells recruited to the lesions expressed PD-1. Classical fractionated whole brain radiotherapy (WBRT) increased the proportion of cytotoxic CD8+ T cells and prolonged survival transiently. Combination therapy of WBRT and PD-1 blockade increased T cells infiltration and prevented the induction of compensatory inhibitory

responses in lymphocytes induced by anti-PD-1 monotherapy. This led to reduced tumor progression and prolonged survival compared to WBRT alone. Analysis of combination treated BrM revealed increased infiltration with PD-L1 positive myeloid cells recruited from the periphery. Impact: We demonstrate that radiotherapy can sensitize low immunogenic BrM to checkpoint blockade. We showed that myeloid cells recruited from the periphery play a crucial role in regulating T cell activation and are implicated in generating an immune suppressive environment, underlying the importance of investigating the role of these cells in acquired resistance to checkpoint blockade in more details.

5/ Thank you for providing "For Further information". Please rename it "For more information". Please also check the URL you provided as the linked page indicated "not found".

6/ As part of the EMBO Publications transparent editorial process initiative (see our Editorial at <http://embomolmed.embopress.org/content/2/9/329>), EMBO Molecular Medicine will publish online a Review Process File (RPF) to accompany accepted manuscripts. This file will be published in conjunction with your paper and will include the anonymous referee reports, your point-by-point response and all pertinent correspondence relating to the manuscript. Let us know whether you agree with the publication of the RPF and as here, or IF YOU WANT TO REMOVE ANY FIGURES from it prior to publication.

I look forward to receiving your revised manuscript.

Yours sincerely,

Lise Roth

Lise Roth, PhD
Editor
EMBO Molecular Medicine

***** Reviewer's comments *****

Referee #1 (Remarks for Author):

The manuscript significantly improved and the most of my points/remarks are addressed. Thus, in the current version the manuscript is suitable for publication. However, the functional evidence of the immune-suppressive role by MDM-TAM is still weak. The most data are descriptive which derive by very elaborated FACS sorting experiments.

Referee #2 (Comments on Novelty/Model System for Author):

The authors have addressed most of the concerns raised by myself and reviewer 1 although mainly by toning down interpretation. This has led to the paper being technically more competent although that comes at the expense of it not saying as much. A few things raised by reviewer 1 cannot be addressed due to the lack of material for them to analyse but I think that their examination of other animals showing the lack of extracranial metastases should be sufficient for the purposes shown here. I believe this is now suitable for publication.

Referee #2 (Remarks for Author):

The authors have done a thorough job of addressing concerns raised by myself and the other reviewers and I think have made the manuscript stronger as a whole.

The responses to my minor concerns were thorough and thoughtful. The change in title and the more complete characterisation of BMDM T cell cocultures leading to a more nuanced appreciation of how these may lead to suppression also addresses my major concerns there. The statistical analysis is now thoroughly detailed addressing my final major concern.

Referee #3 (Comments on Novelty/Model System for Author):

The model system used in this study is well-suited to investigate immune suppression and possible strategies to modulate these responses in metastatic cancer. Furthermore, it will be of interest to those in the tumor field and has high medical impact, especially to investigators in the field of cancer immunotherapy.

Referee #3 (Remarks for Author):

I am satisfied with the revisions to this manuscript and am impressed by the extra data and analyses that were added, especially during a pandemic. I recommend this manuscript to be published in its revised form.

The authors performed the requested editorial changes.

22nd Feb 2021

Dear Dr. Sevenich,

Thank you for sending the revised files. I have looked at everything, and all is fine. I am therefore very pleased to accept your manuscript for publication in EMBO Molecular Medicine!

Your manuscript will be sent to our publisher to be included in the next available issue of EMBO Molecular Medicine.

Please read below for additional important information regarding your article, its publication and the production process.

Congratulations on a nice study!

Yours sincerely,

Lise Roth

Lise Roth, Ph.D
Editor
EMBO Molecular Medicine

Corresponding Author Name: Lisa Sevenich
Journal Submitted to: EMBO Molecular Medicine
Manuscript Number: EMM-2020-13412